# SyMerge: From Non-Interference to Synergistic Merging via Single-Layer Adaptation

## Abstract

Model merging offers an efficient alternative to multi-task learning by combining independently fine-tuned models, but most prior approaches focus mainly on avoiding task interference. We argue instead that the real potential of merging lies in achieving synergy, where tasks enhance one another. Our intuition comes from a pilot study showing that when a classifier trained on one task is paired with the encoder of another, the resulting cross-task performance strongly predicts merge quality. Moreover, adapting even a single task-specific layer can substantially improve this compatibility, suggesting a simple yet powerful lever for synergy. Building on this insight, we introduce SyMerge, a lightweight framework that jointly optimizes one task-specific layer and merging coefficients. To ensure stability without labels, SyMerge employs a robust self-labeling strategy guided by expert model predictions, avoiding the pitfalls of entropy-based adaptation. This minimalist yet principled design achieves state-of-the-art results across vision, dense prediction, and NLP benchmarks, while also producing adapted layers that transfer effectively to other merging methods.

## 1 Introduction

Multi-Task Learning (MTL) (Caruana, 1997) enables models to solve multiple tasks within a single framework by encouraging knowledge transfer. Recently, model merging (Wortsman et al., 2022a; Ilharco et al., 2022; Wortsman et al., 2022b) has emerged as an attractive alternative to conventional MTL (Yu et al., 2020; Misra et al., 2016; Kendall et al., 2018; Hu et al., 2024), as it constructs a multi-task model by directly combining independently fine-tuned models at the parameter level. A central concept is the *task vector* (Ilharco et al., 2023), which represents the weight difference between fine-tuned and pre-trained models, effectively encoding task-specific knowledge and enabling flexible adaptation to new tasks at low cost.

However, simply aggregating task vectors often leads to severe performance degradation due to the various forms of interference (Yadav et al., 2024; Gargiulo et al., 2025; Wang et al., 2024). A line of work (Yang et al., 2024b; Tang et al., 2024b; Yang et al., 2024a; Wei et al., 2025) performed adaptive tuning at test time, moving one step forward beyond simple aggregation by leveraging adaptive mechanisms. By adjusting the merged model in an unsupervised manner at test time, they become significantly robust to distribution shifts; however, they might not explicitly consider interference, so it remains uncertain whether they have indeed resolved it.

To address this challenge more explicitly, previous work explored several mitigation strategies. Some methods tackle interference at the parameter level; these methods employed binary masks to select important weights for each task (Wang et al., 2024) or prune conflicting parameters after assessing their significance and inter-task conflicts (Yadav et al., 2024; Du et al., 2024; Deep et al., 2024). Others analyzed the structural properties of the weights; for instance, leveraging SVD to resolve conflicts within each layer's weight matrix (Gargiulo et al., 2025; Marczak et al., 2025a), or using Fourier analysis to filter interference from low-frequency components (Zheng & Wang, 2025). Some other studies (Ortiz-Jimenez et al., 2023; Jin et al., 2025) can be interpreted as pursuing non-interference through weight disentanglement, where adding one task vector does not impair another. Yet, we believe this might be achievable in practice, and most approaches continue to treat it as the final goal of merging.

In this paper, we argue that the objective of model merging should extend beyond avoiding interference to tackle cross-task interference explicitly. The true potential may lie in achieving synergy, where tasks can actively improve one another. Our motivation comes from a pilot study showing that when a classifier trained on one task is paired with the encoder of another, the resulting cross-task performance strongly predicts merging quality. Moreover, adapting even a single task-specific layer substantially improves this cross-task compatibility. These observations reveal that minimal adaptation can serve as a powerful lever for inducing synergy during merging.

Building on this intuition, we propose **SyMerge**, a lightweight and test-time adaptive framework that jointly optimizes a single task-specific layer and the encoder's merging coefficients. Unlike conventional entropy minimization, which we find unstable, **SyMerge** employs a robust self-labeling strategy guided by expert model predictions. This design yields strong generalization under distribution shifts without introducing extra modules or costly training.

Our contributions are threefold:

- We redefine the goal of model merging from mitigating interference to fostering positive task synergy, moving beyond non-interference or disentanglement.

- We present **SyMerge**, a minimalist yet effective method that adapts only a single layer together with merging coefficients, stabilized by expert-guided self-labeling.

- We demonstrate state-of-the-art performance across vision, dense prediction (segmentation, depth, surface normals), and NLP, and provide analyses showing that the adapted layers are highly transferable and enhance functional alignment across tasks.

## 2 BACKGROUND

### 2.1 PRELIMINARY

**Problem definition.** Multi-Task Learning (MTL) aims to train a single model that performs well across a set of $K$ tasks. Model merging offers an efficient alternative by constructing a multi-task model without costly joint training. Specifically, a pre-trained model $\Theta_{\text{pre}}$ is fine-tuned independently on each task, producing $K$ expert models with weights $\Theta_1, \ldots, \Theta_K$. The key challenge is to design a merging function $\mathcal{M}$ that combines these experts into a unified parameter set $\Theta_{MTL} = \mathcal{M}(\Theta_1, \ldots, \Theta_K)$. The merged model consists of a shared encoder and the original task-specific layers. For task $k$, the model can be written as $f(\cdot; \Theta_k) = f^L(\cdot; \theta_k^L) \circ f^{1:L-1}(\cdot; \Theta_{MTL}^{\text{enc}})$, where $\Theta_{MTL}^{\text{enc}}$ denotes the merged encoder. The goal is to ensure that this single model $f$ achieves strong performance across all $K$ tasks simultaneously.

**Training-free model merging.** These methods determine merging coefficients using predefined heuristics (Yadav et al., 2024; Du et al., 2024) or through costly hyperparameter grid search on a validation set (Wang et al., 2024; Ilharco et al., 2023). While this avoids data-driven optimization on the target distribution, it introduces critical scalability and computational bottlenecks. The requisite grid search becomes computationally prohibitive as evaluation cost scales with the number of tasks, and the memory-intensive element-wise operations often preclude GPU acceleration. A representative approach is *Task Arithmetic* (Ilharco et al., 2023), which captures the task-specific knowledge as the deviation from the pre-trained weights, $\tau_k^l = \theta_k^l - \theta_{\text{pre}}^l$. The merging process is applied to the encoder layers ($l \in \{1, \ldots, L-1\}$), where the merged weights are constructed as $\theta_{\text{MTL}}^l = \theta_{\text{pre}}^l + \lambda \cdot \sum_{k=1}^{K} \tau_k^l$. Here, $\lambda$ is a manually tuned hyperparameter. The final multi-task model is then composed of this single merged encoder and the original task-specific layers from their respective fine-tuned models. However, while simple, their data-agnostic nature incurs suboptimal performance compared to adaptive methods, especially under many-task settings or distribution shifts.

**Test-time adaptive merging.** Another line of methods adapts to the target distribution by optimizing merging coefficients or adapters with unlabeled test data. Since ground-truth labels are unavailable, these methods rely on proxy objectives such as minimizing prediction entropy (Yang et al., 2024b; Tang et al., 2024b). However, entropy-based optimization is limited to classification and does not easily extend to tasks like semantic textual similarity or dense per-pixel prediction. Post-hoc calibration offers an alternative that avoids this issue. It adjusts the feature representations of a merged model using additional adapters, rather than learning the merging coefficients directly (Yang et al.,

2024a; Wei et al., 2025). A milestone work in this line of methods is *AdaMerging* (Yang et al., 2024b). It improves *Task Arithmetic* by constructing a merged model $f(x; \Theta_{\text{MTL}})$ through learning merging coefficients, denoted by $\Lambda = \{\lambda_k^l\}_{k,l}$, in a task- and layer-wise manner. The merged weights are defined as $\theta_{\text{MTL}}^l = \theta_{\text{pre}}^l + \sum_{k=1}^{K} \lambda_k^l \cdot \tau_k^l$. The coefficients $\Lambda$ are learned by optimizing the following objective function: $\min_{\Lambda} \sum_{k=1}^{K} \mathbb{E}_{x \in \mathcal{X}_k^{te}} [\mathcal{L}_k (f(x; \Theta_{\text{MTL}}))]$. Note that ground-truth test labels are not used; it learns coefficients via entropy minimization loss $\mathcal{L}_k$.

Our method builds upon test-time merging approaches. Although these methods may not explicitly account for interference, they are more adaptable to unseen data. Below, we will begin with our motivation for this design choice and describe the directions we aim to achieve moving forward.

## 2.2 MOTIVATION

**Limitations of training-free methods.** Our first motivation arises from the limitation of training-free methods, which are vulnerable when adapting to unseen tasks or domain shifts. To examine whether existing merging methods lack robustness to data quality and distribution shifts, we intentionally corrupt 4 task datasets following Hendrycks & Dietterich (2019) as a prevalent real-world challenge. This setup serves as a rigorous proxy for noisy or out-of-distribution test sets often encountered in wild.

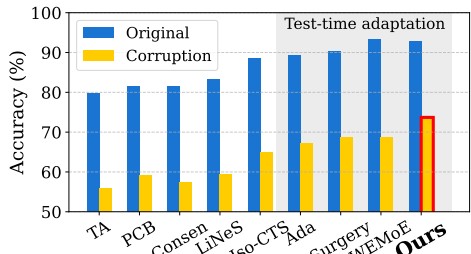

Figure 1: **Training-Free methods collapse under corruption.** Worse than test-time methods on clean data as well, and far more degraded under corruption.

As shown in Figure 1, the performance of training-free methods drops significantly on this corrupted data, revealing their fundamental limitation in handling domain shifts. This failure motivates our approach of leveraging unlabeled test data for robust merging, a principle adapted from test-time adaptation that avoids data leakage. Our empirical results are consistent with prior theoretical work (Xu et al., 2025), which highlights the vulnerability of data-agnostic merging in practice. For more detailed results, see Appendix B.1.

**Rethinking cross-task performance.** Another motivation comes from the intuition that a model's cross-task performance[1] is closely tied to its merging performance. To examine this, we conducted a preliminary study on 20 vision tasks using ViT-B/32. We observed a significant positive correlation ($r = 0.863$, $p < 0.001$) between a model's average cross-task performance and its average merging performance. Specifically, we treat each task in the benchmark as a target task $B$ and iterate through all other tasks $A$ ($A \neq B$). For each pair, we evaluate: (1) the pairwise cross-task performance, and (2) the pairwise merging performance by weight-averaging the two encoders and then evaluating the new merged encoder on $B$'s task. The final metrics for task $B$ are computed by averaging these pairwise values over all source tasks $A$. As shown in Figure 2, a model's ability to generalize across tasks is a strong and reliable predictor of its potential for successful merging. This key insight forms the foundation of our work.

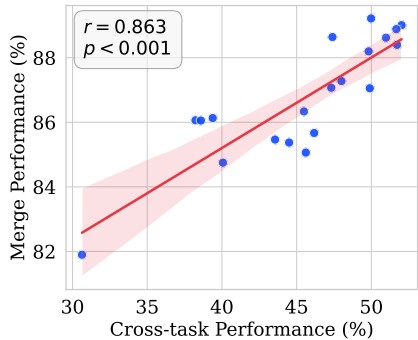

Figure 2: **Cross-task vs. Merge Performance.** Positive correlation observed across 20 vision tasks with regression fit and 95% confidence interval.

Prior work (Ortiz-Jimenez et al., 2023) often regards non-interference as the objective of model merging, where the sole aim is to prevent tasks from degrading one another. However, this perspective inherently imposes a ceiling because cross-task performance cannot go beyond that of the pre-trained model, leaving little room for improvement. We instead highlight the need for positive synergy, where tasks actively enhance one another and collectively achieve performance beyond this ceiling. In Section 3.2, we provide a theoretical analysis showing that improvements in cross-task performance directly translate into more effective merging.

---

[1]We define *cross-task performance* as evaluating model A's encoder with model B's classifier on B's task.

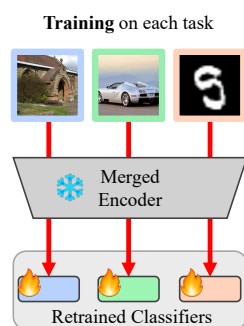
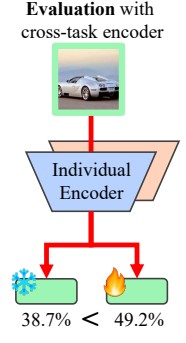
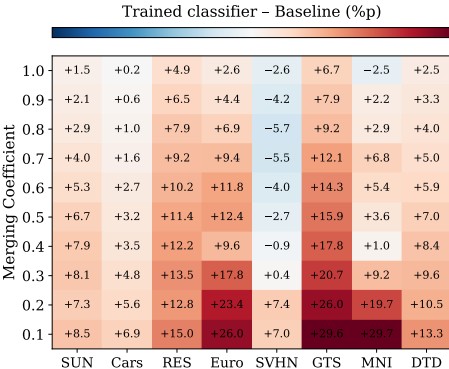

(a) Two-stage protocol for evaluation

(b) Performance gain on 8 cross-tasks

Figure 3: **Two-stage pilot study protocol and its results on 8 cross-tasks using ViT-B/32.** (a) We first enhance a classifier's functional alignment by training it on representations from a general-purpose merged encoder. We then measure this enhancement by evaluating the trained classifier's cross-task performance when paired with the encoder of a different, individual task. (b) The heatmap shows the accuracy gain (%p) over the baseline across 8 tasks (x-axis) under various merged encoder configurations (y-axis, via merging coefficient). The consistently positive gains (red) demonstrate the protocol's effectiveness in enhancing functional alignment.

## 3 METHOD

Our method is motivated by the link between cross-task performance and merge quality. To study this connection, we compared a baseline that pairs a task's original classifier—over-specialized to its native encoder—with an encoder from another task. We then designed a two-stage protocol (Figure 3a): first retraining the task-specific layer $f^L(\cdot; \theta_k^L)$ on features from a fixed merged encoder $f^{1:L-1}(\cdot; \Theta_{\text{MTL}}^{\text{enc}})$ using labeled data $\mathcal{X}_k^{tr}$, and then evaluating the updated layer on an encoder from a different task $m \neq k$. This procedure provides a direct measure of functional alignment (Csiszárik et al., 2021) across tasks.

As shown in Figure 3b, the protocol consistently improved cross-task performance over the baseline (detailed results are in Appendix B.2). The benefit was observed in most cases, although SVHN showed limited or even negative gains, likely due to overlap with MNIST. We also found that the effect is not restricted to the final classifier: adapting a single intermediate block yielded similar improvements (Figure 5). Finally, the degree of improvement depended on the merging coefficients, highlighting the importance of a well-formed merged encoder for enabling stronger alignment.

These observations indicate that adapting just one layer can substantially improve cross-task compatibility, which is a reliable predictor of merge success. However, this protocol relies on ground-truth labels that are unavailable during model merging. The remaining challenge is therefore to achieve the same effect in an unsupervised setting, which is the focus of our proposed method, **SyMerge**.

### 3.1 SYMERGE: SYNERGISTIC MODEL MERGING

**SyMerge** addresses the lack of labels at test time by adopting a self-labeling strategy. A common approach in this setting is to use entropy minimization as a supervisory signal (Yang et al., 2024b). However, this proxy can be unstable and misguide the model, as shown in our analysis (Figure 4) and by prior work (Oh et al., 2025).

To create a more reliable training signal, we use the individually fine-tuned models as expert teachers and adopt a self-labeling strategy (Lee et al., 2013). Our objective is to train the merged model to match the confident predictions of these expert models. We formulate this as minimizing the cross-entropy between the predictions of our merged model and the self-labels from the individual models on the unlabeled test set $\mathcal{X}^{te}$. The objective is defined as $\min\limits_{\{\lambda_k^l\}, \{\theta_k^{\text{tr}}\}} \sum\limits_{k=1}^{K} \mathcal{L}_{CE}\left(C_k^{\text{merged}}, C_k^{\text{ft}}\right)$, where

(i) $C_k^{\text{merged}}$ is the output from our merged model for task $k$; (ii) $C_k^{\text{ft}}$ is the fixed prediction from the corresponding expert model; and (iii) the trainable parameters are the merging coefficients $\{\lambda_k^l\}$ and

a task-specific layer $\{\theta_k^{tr}\}$. Our objective readily extends beyond classification, as the key principle is to mimic the expert's output signal. This is achieved simply by using the appropriate task-specific loss (*e.g.*, L1 loss for regression tasks). Below, we provide an empirical justification for choosing this self-labeling objective over the more common, but less stable, entropy minimization.

Note that a key difference from AdaMerging (Yang et al., 2024b) is that we jointly optimize both the shared encoder's merging coefficients and the task-specific layer. We argue this allows the two components to adapt to each other, leading to better task specialization. We can optionally apply a confidence-based filtering mechanism to further enhance performance (see details in Appendix A).

**On the choice of the objective function.** An effective proxy objective for test-time adaptation must maintain a strong correlation with the ground-truth objective. To verify this, we compute the Spearman correlation[2] on 8 vision tasks with ViT-B/32. A higher correlation indicates that the loss function provides a reliable training signal. We evaluate this correlation for both the initial model merged via Task Arithmetic ("Initial weight") and the final model after it has been optimized with a proxy loss ("Adapted weight"). For the "Entropy", we correlate the entropy of the model's output with the ground-truth loss. In contrast, for "Ours", we correlate our self-labeling loss with the same ground-truth loss.

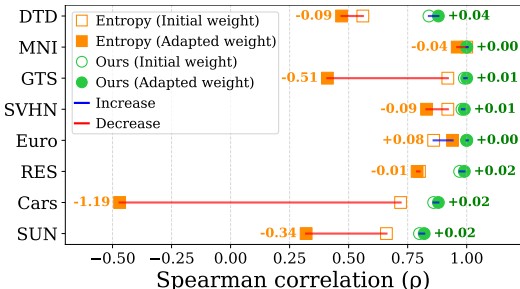

Figure 4: **Spearman correlation of proxy losses with ground truth cross-entropy loss.** We compare coefficients for Entropy and Ours using merged weights before and after training. A coefficient closer to +1 indicates a more reliable proxy for the true objective.

Figure 4 shows that entropy initially correlates moderately with supervised cross-entropy. However, after training, this correlation significantly weakens, and in some cases, entropy diverges sharply (*e.g.*, Cars). This suggests that entropy is not a stable proxy for supervision and may fail to maintain consistency as training progresses, which aligns with prior studies (Yang et al., 2024b; Oh et al., 2025). In contrast, our proposed objective maintains a consistently high correlation, providing a stable and reliable supervisory signal.

### 3.2 THEORETICAL JUSTIFICATION

We theoretically support that stronger cross-task performance leads to better merge performance. When a task vector for task $i$ not only avoids harming task $j$ but actually helps it, the merged model can achieve a strictly lower loss than under the usual non-interference setting.

Let $f(x; \theta_0 + \sum_t \tau_t)$ be a merged model. Following prior work (Ortiz-Jimenez et al., 2023), non-interference (*i.e.*, weight disentanglement) assumes $L_j[f(x; \theta_0 + \tau_i)] = L_j[f(x; \theta_0)]$ for all $i \neq j$. This implies that adding $\tau_i$ leaves task $j$ unchanged, and consequently, the cross-task performance is the same as the pre-trained model. We propose a stronger condition of synergy, where merging task $i$ improves task $j$, $L_j[f(x; \theta_0 + \tau_i)] < L_j[f(x; \theta_0)]$.

**Proposition 1.** *Assume cross-task linearity (Zhou et al., 2024) so that $f(x; \frac{1}{2}(\theta_i + \theta_j)) \approx \frac{1}{2}f(x; \theta_i) + \frac{1}{2}f(x; \theta_j)$, and suppose the loss function is convex in its output. Then the merged model $f_{merge}(x) = f(x; \frac{1}{2}(\theta_i + \theta_j))$ has an expected loss below the case of weight disentanglement.*

The proof in Appendix C shows that this positive cross-task improvement tightens the convex upper bound, giving a clear guarantee of better merging performance. We note that the convexity assumption holds exactly for the cross-entropy loss *with respect to the model outputs* $f(x; \theta)$, and we do not assume convexity in the parameters $\theta$. While cross-task linearity is an approximation in parameter space, our empirical analysis in Appendix D.4 confirms that it holds robustly in our setting. In short, synergistic cross-task effects provide a direct and sufficient path to superior model merging, explaining why our method explicitly seeks higher cross-task performance. For experimental results on scenarios where the cross-task linearity assumption is violated, please refer to Table 5.

---

[2]Spearman correlation quantifies monotonic relationships, commonly used for ranked or non-linear data.

Table 1: **Multi-task performance across 8, 14, 20 vision tasks.** We compare our methods with the competing methods for merging ViT-B/32 and ViT-L/14 fine-tuned models. Our main competitors are the test-time merging methods (bottom group, starting from AdaMerging). All reported results for our methods are the mean and standard deviation computed over 5 runs.

| Method | ViT-B-32 | | | ViT-L-14 | | |
|---|---|---|---|---|---|---|
| | 8 tasks | 14 tasks | 20 tasks | 8 tasks | 14 tasks | 20 tasks |
| Pretrained | 48.0 | 57.1 | 56.0 | 64.5 | 68.1 | 65.1 |
| Individual | 90.5 | 89.5 | 90.4 | 94.2 | 93.2 | 94.0 |
| Task Arithmetic (Ilharco et al., 2023) | 69.1 | 65.4 | 60.8 | 84.5 | 78.4 | 73.9 |
| Ties Merging (Yadav et al., 2024) | 72.9 | 65.2 | 63.1 | 86.0 | 80.5 | 73.0 |
| PCB Merging (Du et al., 2024) | 75.6 | 63.8 | 52.7 | 87.5 | 81.3 | 73.4 |
| LiNeS w/ TA (Wang et al., 2025) | 74.1 | 68.0 | 63.7 | 86.9 | 80.4 | 75.7 |
| Consensus TA (Wang et al., 2024) | 74.9 | 70.3 | 65.0 | 86.6 | 82.3 | 78.9 |
| TSV-M (Gargiulo et al., 2025) | 84.0 | 80.1 | 76.6 | 91.5 | 88.2 | 87.2 |
| ISO-CTS (Marczak et al., 2025a) | 84.2 | 80.6 | 76.9 | 93.0 | 89.7 | 89.2 |
| EMR-Merging (Huang et al., 2024) | 88.7 | 86.1 | 86.6 | 93.7 | 91.4 | 92.0 |
| AdaMerging (Yang et al., 2024b) | 80.1 | 76.7 | 69.6 | 90.8 | 85.2 | 82.1 |
| Surgery w/ Ada. (Yang et al., 2024a) | 87.5 | 84.6 | 84.5 | 92.3 | 90.4 | 89.5 |
| WEMoE (Tang et al., 2024b) | 89.4 | 83.0 | 78.9 | 93.6 | 88.4 | 78.8 |
| ProbSurgery w/ Ada. (Wei et al., 2025) | 87.4 | 84.9 | 84.5 | 92.7 | 90.7 | 90.2 |
| **SyMerge** | **90.1** ± 0.1 | **88.7** ±0.1 | **88.6** ± 0.4 | **94.1** ± 0.0 | **92.8** ± 0.1 | **93.2** ± 0.1 |

## 4 EXPERIMENT

### 4.1 EXPERIMENTAL SETUP

**Datasets and metrics.** We follow the standard setups in Ilharco et al. (2023) and Yang et al. (2024b), using fine-tuned weights on 8 image classification tasks. We extend our experiments to 12 additional tasks, covering a total of 20 tasks as in Wang et al. (2024). All vision tasks are evaluated using accuracy. For dense prediction, we use NYUv2 (Silberman et al., 2012), which contains 1,449 RGB-D indoor scenes. It includes three tasks: 13-class semantic segmentation, depth estimation, and surface normal estimation. We evaluate them with mIoU/pixel accuracy, absolute/relative error, and mean angular error, respectively. For NLP tasks, we adopt the setups from Yu et al. (2024) and Huang et al. (2024), using fine-tuned weights for 8 tasks from the GLUE benchmark (Wang, 2019). Task-specific metrics are employed: Matthews correlation for CoLA (Warstadt et al., 2019), Pearson and Spearman correlations for STS-B (Cer et al., 2017), and accuracy for others.

**Models.** For vision experiments, we employ pre-trained CLIP (Radford et al., 2021) models[3], specifically ViT-B/32, L/14. Most analyses use ViT-B/32. To ensure comparability across 8 tasks, we use publicly available fine-tuned weights[4] from Ilharco et al. (2023). For 14- and 20-task experiments, we fine-tune CLIP ViT-B/32 following the Task Arithmetic configuration. For dense prediction, we adopt ResNet-50 (He et al., 2016) as the backbone, following Tang et al. (2024a). The model is initialized with ImageNet (Deng et al., 2009) pre-trained weights and fine-tuned for each task. For NLP tasks, we use RoBERTa-base (Liu et al., 2019), leveraging publicly available fine-tuned weights from Huang et al. (2024).

### 4.2 MAIN RESULTS

**Merging 8, 14, 20 vision tasks.** Table 1 demonstrates the superiority of our approach, which surpasses all baselines across varying numbers of tasks and model scales. Notably, while the performance of competing methods degrades sharply with more tasks, **SyMerge** shows remarkable consistency, closely approaching the performance of the individual models that serve as the upper bound. Furthermore, it establishes a significant margin over other test-time adaptive methods. The per-task performance can be found in Appendix B.3.

---

[3] https://github.com/mlfoundations/open_clip
[4] https://github.com/mlfoundations/task_vectors

Table 2: **Multi-task performance across three dense prediction tasks.** We compare the performance of merging ResNet50 models fine-tuned on three dense prediction tasks in NYUv2.

| Method | Segmentation | | Depth Estimation | | Normal |
|---|---|---|---|---|---|
| | mIoU↑ | Pix Acc↑ | Abs Err↓ | Rel Err↓ | Mean↓ |
| Individual | 52.0 | 74.2 | 41.5 | 17.3 | 24.2 |
| Weight Averaging | 36.6 | 64.0 | 55.0 | 23.2 | 30.0 |
| Task Arithmetic (Ilharco et al., 2023) | 31.6 | 60.3 | 56.7 | 24.0 | 30.6 |
| Ties-Merging (Yadav et al., 2024) | 39.9 | 62.7 | 61.3 | 27.3 | 36.2 |
| MagMax (Marczak et al., 2025b) | 24.7 | 54.7 | 60.3 | 23.9 | 30.3 |
| LiNeS w/ TA (Wang et al., 2025) | 36.2 | 64.0 | 54.2 | 22.4 | 29.1 |
| EMR-Merging (Huang et al., 2024) | 41.5 | 67.2 | 48.6 | 19.4 | 26.5 |
| Surgery w/ TA (Yang et al., 2024a) | 43.3 | 67.4 | 55.3 | 24.7 | 34.7 |
| ProbSurgery w/ TA (Wei et al., 2025) | 43.6 | 67.6 | 52.6 | 22.3 | 36.7 |
| **SyMerge** | **49.8** $_{\pm 0.3}$ | **73.1** $_{\pm 0.2}$ | **45.3** $_{\pm 0.6}$ | **18.8** $_{\pm 0.5}$ | **26.2** $_{\pm 0.1}$ |

Table 3: **Multi-task performance across 8 NLP tasks.** We present a performance comparison of merging the RoBERTa models fine-tuned on 8 tasks in GLUE.

| Method | CoLA | SST2 | MRPC | STSB | QQP | MNLI | QNLI | RTE | Avg. |
|---|---|---|---|---|---|---|---|---|---|
| Individual | 60.2 | 94.0 | 89.2 | 90.6 | 91.4 | 87.2 | 92.7 | 79.1 | 85.6 |
| Weight Averaging | 14.0 | 64.1 | 69.4 | 31.8 | 75.4 | 42.2 | 58.7 | 55.2 | 51.3 |
| Task Arithmetic | 18.8 | 85.9 | 79.9 | 74.0 | 83.8 | 59.1 | 69.7 | 62.1 | 66.7 |
| MagMax | 17.3 | 76.0 | 70.8 | 71.3 | 85.8 | 70.4 | 59.5 | 45.1 | 62.0 |
| Ties-Merging | 20.5 | 84.4 | 81.1 | 58.2 | 85.7 | 64.7 | 74.8 | 43.0 | 64.0 |
| LiNeS | 26.1 | 86.4 | 78.9 | 72.9 | 83.3 | 56.0 | 75.9 | 59.9 | 67.4 |
| EMR-Merging | 40.0 | 93.4 | 86.3 | 82.8 | **89.7** | **85.5** | **89.6** | 74.4 | 80.2 |
| Surgery w/TA | 46.8 | 93.6 | **89.7** | 88.7 | 85.1 | 78.0 | 85.3 | 76.5 | 80.5 |
| ProbSurgery w/TA | 54.7 | 93.6 | **89.7** | 89.4 | 85.2 | 77.6 | 85.3 | 76.9 | 81.6 |
| **SyMerge** | **60.0**$_{\pm 0.1}$ | **93.8**$_{\pm 0.3}$ | 89.2$_{\pm 0.0}$ | **90.4**$_{\pm 0.2}$ | 85.2$_{\pm 1.1}$ | 84.0$_{\pm 0.4}$ | 89.2$_{\pm 0.5}$ | **79.1**$_{\pm 0.0}$ | **83.9**$_{\pm 0.2}$ |

**Merging dense prediction tasks.** Dense prediction tasks, such as segmentation, depth estimation, and normal estimation, pose significant challenges due to their inter-task heterogeneity; for instance, semantic segmentation relies on high-level semantic abstraction, whereas depth and surface normal estimation depend on low-level geometric features. Despite this, most model merging studies focus on classification, leaving these high-interference scenarios largely unexplored. Table 2 presents results for these tasks, highlighting the limitations of previous methods. ProbSurgery, which performs comparably to individual models on 8 classification tasks, suffers substantial performance drops in depth and normal estimation. Similarly, EMR-Merging, despite leveraging task-specific dynamic masks, struggles to maintain segmentation performance. In contrast, our approach effectively balances these tasks, maintaining performance close to individual models even in dense prediction.

**Merging 8 NLP tasks.** As shown in Table 3, our **SyMerge** largely improved over the best method on average. A key observation lies in the results on CoLA, a task evaluating grammatical acceptability (syntactic), which inherently differs from other GLUE tasks that focus on semantic understanding. Due to this task heterogeneity, distillation-based methods (*e.g.*, Surgery, ProbSurgery) and EMR-Merging suffer from a significant performance drop on CoLA, **SyMerge** demonstrates consistent performance close to individual models across all tasks. These results highlight our method's robustness, as it not only maximizes average performance but also prevents significant degradation on any single task. This shows **SyMerge**'s superior ability to stably integrate a diverse set of tasks.

## 4.3 EMPIRICAL ANALYSES

**Transferable cross-task ability.** Our work argues that enhancing functional alignment is key to successful merging, and our pilot study showed that adapting a single layer is an effective way to achieve this. We now investigate if the adapted layer learns a truly transferable capability, *i.e.*, if it can be reused as a plug-and-play component to improve other merging methods. To test this,

Table 4: **Cross-task transferability check.** Classifiers trained with our method replace the original zero-shot classifiers (in a pre-trained model) connected to merged encoders (Task Arithmetic and AdaMerging) for evaluating different tasks. This replacement yields substantial performance gains on both merged and cross-task evaluations without training on target tasks, demonstrating the high transferability and improved functional alignment.

| Encoder | Classifier | Merged | Cross |
|---------|-----------|--------|-------|
| TA | Zero-shot | 69.1 | 49.8 |
|    | Ours | **79.6** (+10.5) | **54.8** (+5.0) |
| Ada | Zero-shot | 80.1 | 50.1 |
|     | Ours | **87.7** (+7.6) | **54.1** (+4.0) |

Table 5: **Merging models with different initializations.** We merge EuroSAT (DataComp-ViT) and DTD (OpenAI-ViT), which originate from disjoint basins. While standard methods collapse due to severe loss barriers, `SyMerge` uniquely recovers performance, showing its capability to bridge the gap where linear connectivity fails.

| Method | EuroSAT | DTD | Avg. |
|--------|---------|-----|------|
| Individual (EuroSAT) | 98.1 | 3.6 | 50.8 |
| Individual (DTD) | 2.0 | 79.4 | 40.7 |
| Weight Averaging | 11.6 | 2.2 | 6.9 |
| AdaMerging | 8.6 | 2.1 | 5.4 |
| Surgery | 38.1 | 13.1 | 25.6 |
| **Ours** | **96.2** | **62.0** | **79.1** |

we use the merged encoders from two leading baselines, Task Arithmetic (Ilharco et al., 2023) and AdaMerging (Yang et al., 2024b), as fixed backbones. We then replace their original zero-shot classification heads with the corresponding classifiers trained by our `SyMerge` process and evaluate the new model combinations without any further training.

In Table 4, we assess this transferability using two metrics. "Merged" indicates the accuracy of the final multi-task model, which combines a baseline's merged encoder (*i.e.*, $\theta_{pre}^l + \sum_k \lambda_k^l \cdot \tau_k^l$) with the classifier. "Cross", on the other hand, quantifies average generalization by pairing an encoder built from a single scaled task vector for task $i$ (*i.e.*, $\theta_{pre}^l + \lambda_i^l \cdot \tau_i^l$) with a classifier from a different task $j$ (where $i \neq j$) and averaging the results over all such pairs.

The results show a remarkable degree of transferability. When paired with the TA encoder, it boosts merged performance by 10.5%p and, critically, improves cross-task performance by 5.0%p. We observe similar substantial gains with the AdaMerging encoder. These significant gains demonstrate that the layer trained by `SyMerge` learns a robust and general function not overfitted to its original encoder. This provides strong evidence that our method effectively enhances functional alignment and produces highly transferable components.

**Improvement beyond classifier adaptation.** One might suspect that our performance gains stem primarily from fine-tuning the task-specific layer rather than improving the merged encoder representations. To investigate this, we conducted an ablation using ViT-B/32 on 8 tasks where we retained the merged encoder learned by `SyMerge` but replaced the fine-tuned classifier with the original zero-shot classifier. Remarkably, this configuration achieves an average accuracy of 82.4%, representing a substantial improvement over the standard Task Arithmetic baseline with the same zero-shot classifier (69.1%). This confirms that `SyMerge` significantly enhances the quality of the underlying merged encoder itself, independent of the classifier adaptation.

**Merging across disjoint basins.** Our theoretical analysis assumes that models reside within a shared optimization basin (i.e., cross-task linearity holds). To rigorously test `SyMerge` under conditions where this assumption is violated, we conducted a stress test by merging models derived from *different initializations*, effectively forcing the merge across disjoint loss basins where a significant loss barrier exists. We merged a DTD model fine-tuned from OpenAI-ViT and a EuroSAT model from DataComp-ViT. Since standard Task Arithmetic is inapplicable due to the lack of a common pre-trained weight, we defined task vectors relative to the simple weight average of the two models.

Table 5 illustrates a catastrophic failure for standard methods. Weight Averaging collapses (6.9%), confirming the presence of a severe permutation misalignment and loss barrier between the two models. Surgery also fails to recover performance (25.6%), as external adapters cannot rectify a fundamentally misaligned backbone. In contrast, `SyMerge` successfully bridges this gap, recovering the average accuracy to 79.1%. This result indicates that `SyMerge` is not merely applying distillation; it provides a practical solution for integrating models with severe functional or parameter-level discrepancies that cannot be resolved by existing merging methods. By jointly optimizing the merging coefficients and the internal layer, `SyMerge` effectively realigns the conflicting representations even when the parameter-space linearity breaks down.

Table 6: **Ablation on optimization components.** Layer-only adaptation suffers from scalability issues (dropping to 75.0% on 20 tasks), while coefficient optimization is stable but limited by unaligned heads. **SyMerge** jointly optimizes both, effectively resolving this trade-off to achieve superior performance and robustness.

| Trainable | | Avg. on $N$ Tasks | | |
|---|---|---|---|---|
| Coef | Layer | $N$=8 | $N$=14 | $N$=20 |
| ✓ | | 84.6 | 82.4 | 81.6 |
| | ✓ | 88.2 | 83.0 | 75.0 |
| ✓ | ✓ | **90.1** | **88.7** | **88.6** |

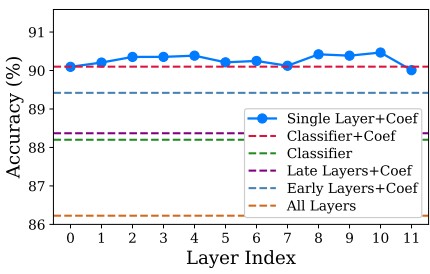

Figure 5: **Impact of training task-specific layers for task synergistic merging.** We compare partial trainings defined by specific layers with the coefficient ("Coef") training. Single-layer/classifier training (with coefficients) works; multi-layer training fails.

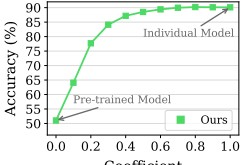
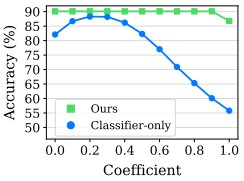

(a) Impact of supervisory model

(b) Robustness to initialization

Figure 6: **More analyses with merging coefficients.** (a) Refined supervisory models merged under various coefficients are tested. Our design choice – using the unmerged individual models – performs near-optimally. (b) Our method shows strong robustness to different initial merging coefficients.

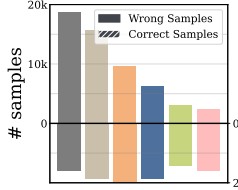
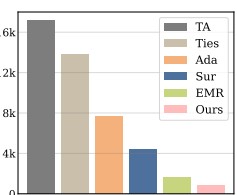

(a) Prediction discrepancy

(b) Overall difference

Figure 7: **Prediction discrepancies between merged and individual model.** (a) The upper bars indicate predictions correctly classified by individual models but misclassified by the merged model; the lower (hatched) bars indicate the opposite. (b) Represents the overall difference.

**Component analysis.** We ablate our joint optimization strategy to confirm that both the merging coefficients and the task-specific layer are necessary for optimal performance. In Table 6, the results confirm that jointly optimizing both yields a significant synergistic gain, consistently outperforming the optimization of either component alone. The analysis reveals a key trade-off: layer-only adaptation excels on 8 tasks but scales poorly as task interference grows in the fixed shared encoder. Conversely, coefficient-only optimization is more robust against an increasing number of tasks but is ultimately limited by a sub-optimal, unaligned task-specific layer. This proves the two components are highly complementary. **SyMerge**'s joint optimization refines the shared encoder while concurrently adapting the task-specific layer to its merged representations, creating the most robust and scalable strategy.

**Effect of adjusting different layers.** A natural question is whether the improvements observed in **SyMerge** are specific to training the classifier. To explore this, we analyze the effect of training different layers beyond the classifier by updating merging coefficients along with each of the 12 transformer layers in the merged encoder. As shown in Figure 5, training a single internal layer achieves accuracy comparable to training the classifier, suggesting that task-specific information can be effectively captured by refining a single layer rather than modifying the entire model. Identifying the optimal layer could further enhance performance. On the other hand, training multiple layers at once, either from the early stage (layers 1–6) or the later stage (layers 7–12), leads to a performance drop. This suggests that updating too many layers disrupts the task-agnostic knowledge embedded in the pre-trained model, reducing its ability to generalize across tasks. See Table C for details.

**More studies with merging coefficients.** We study whether employing individual model predictions without any refinement is a sensible choice for supervisory signals. We refine predictions by merging individual models with a pre-trained model, which is expected to yield more precise predictions. Specifically, we handle each individual model as a pre-trained model with an added task vector scaled by a merging coefficient, ranging from 0 (pre-trained model) to 1 (individual model). We experiment across these settings, and the results are shown in Figure 6a, where performance

consistently improved as predictions approached the individual model. This indicates that guiding the classifier with a task-agnostic pre-trained model limits its ability to capture task-specific details effectively. Interestingly, a combined model (at the coefficient 0.8) performs the best, which is about 90.2%. Finding the optimal coefficient would require cumbersome hyperparameter tuning, so using the existing individual models directly is a practical option. Efficiently searching for the coefficient would be a potential research direction in the future. Also, our method benefits from robust merging coefficient initialization. To highlight this, we present training curves under different initialization settings. As shown in Figure 6b, when only the classifier is trained (*i.e.*, fixed shared encoder), the performance is sensitive to initialization. However, training both the classifier and coefficients together improves robustness.

**Prediction discrepancy analysis.** To evaluate the impact of our approach, we analyze prediction discrepancies between the merged model and the individual models, which often serve as a performance upper bound in multi-task learning (MTL). Figure 7a visualizes this by distinguishing merging-induced errors (upper bars: merged model fails, individual succeeds) from generalization gains (lower bars: the reverse).

Among the evaluated approaches, our method achieves the smallest prediction discrepancy, significantly reducing the number of misclassified samples. Figure 7b further summarizes the overall impact by computing the net difference between the upper and lower bars. A smaller value in this metric indicates a more balanced model that preserves individual model accuracy while capturing additional gains. The proposed method consistently outperforms alternatives, demonstrating its effectiveness in minimizing errors while improving generalization, leading to enhanced MTL performance.

## 5 CONCLUSION

In this work, we introduce `SyMerge`, an effective method that redefines the goal of model merging from mitigating interference to creating synergy, where tasks mutually enhance one another. `SyMerge` achieves this via a lightweight, test-time adaptive process: it jointly optimizes task-vector coefficients and a single task-specific layer on unlabeled data, using a robust self-labeling strategy for stable supervision. Our experiments confirm the broad effectiveness of `SyMerge`, which sets a new state-of-the-art across diverse benchmarks, including vision, dense prediction, and NLP, while scaling robustly as the number of tasks increases. We validate our core insight, that this process enhances functional alignment, by showing our adapted layers are highly transferable and significantly boost other merging methods. Due to its strong performance and compatibility with existing methods, `SyMerge` offers a practical solution for building powerful, scalable multi-task models.

**Limitation.** While our self-labeling method using expert predictions is empirically more stable than entropy-based approaches, its performance is ultimately constrained by the quality of the expert models. However, our observation in Figure 6a that a merged model can yield a slight performance advantage suggests this dependency is not a hard ceiling. This motivates a promising future direction of dynamically combining individual models to compose a more robust expert.

### REPRODUCIBILITY STATEMENT

Our experiments are conducted using PyTorch 1.12.1. Most results are obtained on an NVIDIA RTX 4090 GPU, while experiments involving ViT-L/14 are performed on an NVIDIA RTX A6000 GPU. Experimental setup and hyperparameter configurations are provided in the Appendix. The code will be released publicly upon publication of the paper.

### THE USE OF LARGE LANGUAGE MODELS

During the preparation of this manuscript, a large language model (LLM) is used as an assistive tool to improve grammar and clarity. The role of these tools is strictly limited to proofreading and refining the text composed by the authors. We affirm that the LLMs are not used for core research aspects, including the generation of ideas, experimental design, data analysis, or the interpretation of results. The authors bear full responsibility for the entire content of this paper.

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

# Appendix

This Appendix provides an overview of our experimental setup and baselines, further experimental results across tasks, a theoretical proof, and additional empirical analyses. The detailed descriptions of each section are summarized as follows:

- Appendix A: Detailed experimental setup and hyperparameter configurations;
- Appendix B: Additional results for motivation, pilot study, vision, and NLP tasks;
- Appendix C: Proof for Proposition 1;
- Appendix D: Further analyses on task-specific layers, loss functions, convergence, and sparsity.

## A    EXPERIMENTAL SETUP

### A.1    TRAINING DETAILS

**Main experiments.** We initialize the layer-wise merging coefficients for all layers to 0.3 by default, following the values used in the previous method (Yang et al., 2024b). However, for the 14 and 20 vision tasks, the coefficients are initially set to 0.1, as using 0.3 incurred significantly lower Task Arithmetic (Ilharco et al., 2023) performance for these tasks, presumably due to the increased number of tasks. We use the Adam (Kingma, 2015) optimizer with momentum parameters (0.9, 0.999) to update the coefficients and the classifier. For vision tasks, the learning rate is set to 0.01 when training the classifier and 0.001 when training only the merging coefficients. For dense prediction tasks, the learning rate of 0.0001 is used. For NLP tasks, the learning rate of 0.0005 is used, regardless of whether the classifier, the merging coefficients, or both are being updated. Consistent with prior works (Yang et al., 2024b; Tang et al., 2024; Yang et al., 2024a), vision tasks are trained over 500 iterations, using a batch size of 32. Dense prediction tasks are trained for 25 iterations with a batch size of 16. NLP tasks are trained over 1,000 iterations with a batch size of 32. For each task, the loss is computed, and the model parameters are updated immediately following the forward pass for each batch. To enable sequential processing, the order of tasks is randomized. To ensure a fair evaluation, the number of samples per task and all other experimental settings are identically applied to the other test-time adaptive model merging methods.

**Discussions on implementation.** Previous methods (Yang et al., 2024b; Tang et al., 2024b) sample a batch for each task, pass each batch through a shared backbone, and then process it through the corresponding task-specific head, which is typical. The losses from each task are then combined, and a single gradient update is performed after completing the forward pass for all tasks. However, this approach becomes less efficient as the number of tasks or model size increases.

To address this, we experiment with a sequential update strategy, where updates are applied immediately after each task's forward pass instead of the typical one. Interestingly, this sequential update approach gives a positive byproduct: performance improvements over the traditional approach. While the original AdaMerging achieves an average accuracy of 80.1%, our sequential update strategy improves it to 81.6% for the 8 vision tasks using ViT-B/32. Similarly, our method also enjoys a gain when using sequential updates. We argue that this improvement may result from mitigating catastrophic forgetting in continual learning (Kirkpatrick et al., 2017) by training tasks sequentially rather than simultaneously.

For vision tasks, we employ a confidence-based filtering strategy to ensure the model learns from reliable supervisory signals. As filtering uncertain cases is known to enhance model robustness (Oh et al., 2024; 2025), we adopt a relative confidence-based approach instead of using a challenging fixed threshold. Specifically, we exclude samples where the merged model's top-1 confidence is higher than that of the individual models. This ensures the merged model is supervised by more reliable data, leading to more effective adaptation and improved performance.

**Pilot study.** The aforementioned training details are applied almost identically to the experiments in the pilot study of our main paper. As stated in the main paper, only the classifier was chosen (as a straightforward and practical option) to train on the given Task Arithmetic merged weights for the 8 vision tasks using ViT-B/32. A difference is that, instead of using the test dataset, the training dataset is used in a supervised manner to produce the results closer to the upper bound, in which the difference stands out more. We train the models for only one epoch.

## A.2 BASELINE DETAILS

We compare our method against a diverse set of baselines, ranging from simple merging strategies to advanced methods that employ different mechanisms to address conflicts between tasks.

**Pretrained** indicates a model that predicts multiple tasks without additional fine-tuning for task-specific requirements. However, the absence of task-specific information for downstream tasks generally leads to poor performance.

**Individual** refers to the fine-tuning of individual pre-trained models for each task. Since there is no interference between tasks, it has been regarded as the upper bound of task-specific performance.

**Weight Averaging** merges multiple individual models by directly averaging their parameters to create a single model for multi-task learning. Although simple, this method lacks task-specific adjustments.

**Task Arithmetic** (Ilharco et al., 2023) defines the difference between the fine-tuned and pre-trained model parameters as a task vector. By combining multiple task vectors and adding them to the pre-trained model, it enables multi-task learning.

**MagMax** (Marczak et al., 2025b) merges task vectors (Ilharco et al., 2023) by selecting the parameter with the largest magnitude for each position, consolidating knowledge into a single model without retaining task-specific data.

**Ties Merging** (Yadav et al., 2024) highlights the importance of addressing interference in task arithmetic-based merging. It involves removing redundant parameters from the task vector and resolving parameter sign conflicts.

**PCB-Merging** (Du et al., 2024) resolves conflicting parameter values that arise during model merging by using intra-task importance and inter-task similarity to rescale and prune task vectors.

**LiNeS** (Wang et al., 2025) observes that reducing the influence of shallow layers helps prevent distortion of general representations, thereby simplifying merging coefficient selection by allowing them to increase linearly with layer depth.

**Consensus TA** (Wang et al., 2024) enhances Task Arithmetic (Ilharco et al., 2023) by using Consensus Merging, which retains only weights beneficial to multiple tasks while eliminating irrelevant or task-specific weights. This process uses task-specific binary masks to identify relevant weights and forms a consensus mask to minimize task interference.

**TSV-M** (Gargiulo et al., 2025) decomposes per-layer task matrices using Singular Value Decomposition (SVD) to obtain low-rank Task Singular Vectors (TSVs). It then decorrelates them to mitigate interference when merging models

**ISO-CTS** (Marczak et al., 2025a) improves alignment by flattening the singular value spectrum of the merged task matrix, creating a uniform common subspace. To better preserve unique features, it further enhances this common subspace by incorporating task-specific singular vectors.

**EMR-Merging** (Huang et al., 2024) involves three steps: Elect, Mask, and Rescale-Merging. These steps select key parameters to form a unified model, apply task-specific masks for each task, and adjust scales to achieve better performance.

**AdaMerging** (Yang et al., 2024b) adaptively learns merging coefficients at the task or layer level by minimizing the entropy of predictions on unlabeled test data.

**Representation Surgery** (Yang et al., 2024a) reduces representation bias by training a task-specific module that aligns the merged model's features with those of the individual models.

**WEMoE** (Tang et al., 2024b) utilizes a Mixture of Experts (MoE) module to dynamically separate and integrate shared and task-specific knowledge based on input samples. By training the router on unlabeled test data, it optimizes routing weights and improves task-specific performance.

**ProbSurgery** (Wei et al., 2025) mitigates representation bias by modeling the bias as a learnable distribution to better capture the uncertainty that arises from parameter interference.

Table A: Performance comparison on the original test set versus the average of corrupted test sets.

| Method | Original Test Set | | | | | Corrupted Test Set (Average) | | | | |
|---|---|---|---|---|---|---|---|---|---|---|
| | Cars | EuroSAT | RESISC45 | GTSRB | Avg. | Cars | EuroSAT | RESISC45 | GTSRB | Avg. |
| Pretrained | 59.6 | 45.0 | 60.2 | 32.6 | 49.4 | 54.1 | 17.7 | 49.9 | 20.9 | 35.6 |
| Individual | 77.7 | 99.9 | 96.1 | 98.7 | 93.1 | 73.0 | 54.1 | 84.9 | 86.2 | 74.5 |
| Task Arithmetic | 67.0 | 94.0 | 82.6 | 75.1 | 79.7 | 61.3 | 46.6 | 68.6 | 47.0 | 55.9 |
| Ties-Merging | 67.5 | 83.7 | 79.3 | 65.4 | 74.0 | 62.6 | 41.3 | 67.1 | 39.1 | 52.5 |
| PCB-Merging | 70.8 | 89.6 | 84.6 | 80.7 | 81.4 | 65.3 | 46.0 | 72.1 | 52.9 | 59.1 |
| LiNeS | 69.5 | 96.3 | 84.6 | 82.4 | 83.2 | 64.1 | 50.1 | 70.4 | 53.1 | 59.4 |
| Consensus TA | 67.1 | 95.4 | 78.0 | 84.9 | 81.4 | 61.4 | 50.1 | 48.0 | 70.2 | 57.4 |
| Iso-CTS | 73.6 | 96.3 | 91.3 | 93.4 | 88.6 | 67.5 | 51.4 | 64.2 | 76.9 | 65.0 |
| AdaMerging | 76.2 | 96.0 | 87.5 | 97.0 | 89.2 | 68.5 | 44.4 | 74.6 | 81.4 | 67.2 |
| Surgery w/ Ada | 73.4 | 98.5 | 91.2 | 97.8 | 90.2 | 67.2 | 52.4 | 77.8 | 78.0 | 68.8 |
| WEMoE | 79.1 | 99.3 | 95.5 | 99.1 | 93.2 | 74.3 | 43.3 | 86.9 | 70.1 | 68.6 |
| **SyMerge** | 77.6 | 99.4 | 95.3 | 98.3 | 92.7 | 72.9 | 53.1 | 84.3 | 84.6 | 73.7 |

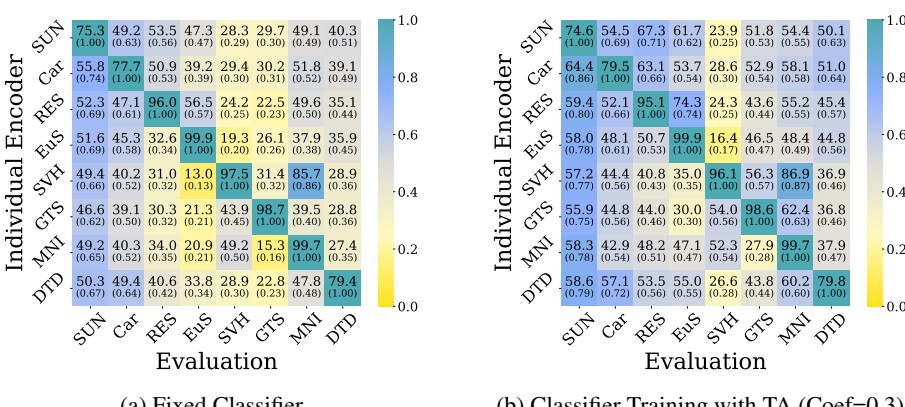

(a) Fixed Classifier      (b) Classifier Training with TA (Coef=0.3)

Figure A: **Cross-task evaluations across diverse encoders and heads from different tasks.** Each cell $(i, j)$ displays the accuracy when features from task $i$'s visual encoder are classified by task $j$'s classifier. The upper value in a cell shows the absolute task accuracy, and the number below (in parentheses) denotes the relative performance compared to the diagonal. (a) Base (untrained) classifiers generally deteriorate when evaluated on other tasks or using different encoders. (b) Classifiers trained with Task Arithmetic (TA) features show mostly reduced performance loss (*i.e.*, higher relative numbers). This suggests that adjusting task-specific layers effectively achieves better functional alignment.

## B MORE EXPERIMENTAL RESULTS

### B.1 EXPERIMENTS ON CORRUPTED TEST DATASETS

To evaluate the robustness of merging methods against distribution shifts, we conduct experiments on corrupted datasets. Following the protocol of ImageNet-C (Hendrycks & Dietterich, 2019), we apply 7 distinct types of corruption (*e.g.*, motion blur, impulse noise, gaussian noise, pixelate, spatter, contrast, and JPEG compression) at severity level 5 to the test sets of 4 vision tasks (Cars, EuroSAT, RESISC45, GTSRB). This setup provides a rigorous testbed for evaluating performance on out-of-distribution data. In the main paper, we presented the average performance across various corruption types and tasks to demonstrate the overall vulnerability of training-free methods to distribution shifts (Figure 1). This section provides a comprehensive breakdown of those results.

Table A shows the average performance across all 7 corruption types, revealing specific vulnerabilities in baseline methods. For instance, WeMoE and Adamerging exhibit a significant performance drop on the EuroSAT task, while some training-free methods are particularly susceptible to GTSRB. In contrast, our method maintains significantly higher accuracy across all tasks, demonstrating superior robustness. For a more granular analysis, Table H details the performance of each merging method on all 4 tasks, individually evaluated across each of the 7 corruption types.

## B.2 Pilot Study

Figure Aa shows cross-task evaluations via individual visual encoders with their respective base classifiers; Figure Ab shows results from combining individual visual encoders with trained classifiers from Task Arithmetic. We observe consistent improvements in cross-task performance when using the classifiers trained on other tasks. For instance, integrating the GTSRB (Stallkamp et al., 2011) classifier trained on Task Arithmetic features with the SVHN (Netzer et al., 2011) encoder achieves 56.3% accuracy (Figure Ab), surpassing the base classifier's 31.4% (Figure Aa). These results suggest that training the classifier on merged features for the GTSRB task also facilitates the alignment of individual SVHN features with the GTSRB task. This demonstrates the trained classifier's enhanced ability to achieve functional alignment with the encoder. Figure I presents the cross-task evaluation results of a classifier trained on a Task-Arithmetic merged encoder with merging coefficients ranging from 0.1 to 1.0, showing the task-pair results corresponding to the main paper's Figure 3b. From this, we can observe a clear variance in performance depending on the choice of the merging coefficient. This suggests that even when task-specific layers are trained, the choice of a shared encoder can lead to suboptimal performance.

## B.3 Vision Tasks

We reported the average performance of ViT-B-32 and ViT-L-14 on 8, 14, and 20 tasks in Table 1 of the main paper. In Tables I, J, K, L, M, and N, we present the per-task performance of ViT-B-32 and ViT-L-14 for the 8-, 14-, and 20-task settings. For each task, our method consistently outperforms other approaches and achieves performance close to that of individual models.

## B.4 NLP Tasks

We conduct experiments to investigate how the training of the classifier and merging coefficients, respectively, affects the performance of NLP tasks in **SyMerge**, as presented in Table 3 of the main paper. Figure B shows the results of merging RoBERTa (Liu et al., 2019) models for 8 NLP tasks using the predictions of individual models as guidance, optimized through the cross-entropy loss. As mentioned in the main paper, evaluation metrics differ across tasks: the Matthews correlation coefficient is used for CoLA (Warstadt et al., 2019), the average of Pearson and Spearman correlations is applied to STS-B (Cer et al., 2017), and the accuracy is used for all other tasks (Socher et al., 2013; Dolan & Brockett, 2005; Iyer et al., 2017; Williams et al., 2018; Rajpurkar et al., 2016; Giampiccolo et al., 2007). The approach where both the classifier and coefficients are trained simultaneously corresponds to **SyMerge**. To examine the role of classifier training in NLP tasks, ablation experiments are conducted by removing specific components. The results show that training only the coefficients leads to the lowest performance while training only the classifier achieves a relatively high performance. Furthermore, training both the classifier and coefficients together demonstrated a complementary effect, achieving the highest performance.

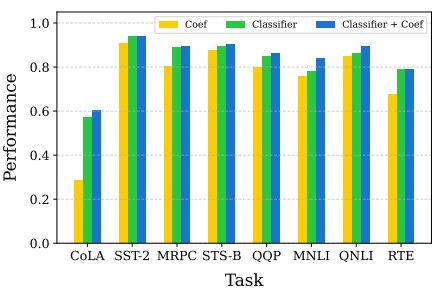

Figure B: **Multi-task performance when merging RoBERTa models on 8 tasks.** Yellow indicates training only coefficients, green indicates training only the classifier, and blue indicates training both in our method.

## B.5 Robustness to Data Scarcity

In real-world deployments, models often face a stream of queries or strictly limited data access rather than a large, pre-collected batch. To evaluate **SyMerge** in such data-scarce and practical scenarios, we conduct experiments in two distinct settings: *Online Test-Time Adaptation* and *Limited-Data Adaptation*.

**Online Adaptation.** First, we simulate a streaming setting where the model receives unlabeled test samples sequentially (batch size of 1). For each incoming sample, the model updates its parameters and is immediately evaluated on that sample, prohibiting multiple passes. Table Ba compares the

Table B: **Robustness to Data Scarcity.** We evaluate `SyMerge` in (a) Online TTA (streaming) and (b) Limited-Data scenarios on ViT-B/32.

(a) Online TTA Performance.

| Method | Seen Samples | | | | | |
|--------|------|------|------|------|------|------|
|  | 10 | 20 | 50 | 100 | 200 | 500 |
| AdaMerging | 70.0 | 71.5 | 71.1 | 72.0 | 72.6 | 74.1 |
| Surgery | 70.0 | 71.1 | 70.9 | 73.4 | 73.2 | 73.6 |
| **Ours** | **73.2** | **75.3** | **75.7** | **77.1** | **80.0** | **82.1** |

(b) Data Efficiency.

| Method | Unlabeled Data Ratio | | | | |
|--------|-----|-----|------|------|------|
|  | 1% | 5% | 10% | 20% | 100% |
| AdaMerging | 76.0 | 77.2 | 77.5 | 77.7 | 80.1 |
| Surgery | 72.8 | 77.3 | 78.4 | 78.9 | 80.9 |
| **Ours** | **78.1** | **84.5** | **86.7** | **88.0** | **90.1** |

cumulative average performance of `SyMerge` against leading test-time adaptation methods on ViT-B/32. `SyMerge` demonstrates remarkable data efficiency. Specifically, in terms of *rapid adaptation*, with only 10 samples seen, `SyMerge` achieves 73.2% accuracy, surpassing AdaMerging and Surgery by over 3.0%p. Furthermore, regarding *sustained improvement*, as the data stream continues to 500 samples, `SyMerge` continuously improves to 82.1%, establishing a significant margin (∼8.0%p) over the baselines, which tend to plateau early.

**Limited-Data Efficiency.** We further evaluated performance using varying amounts of unlabeled test data (ranging from 1% to 100% of the test set). As shown in Table Bb, `SyMerge` consistently outperforms optimization-based TTA methods (AdaMerging, Surgery) as well as training-free merging baselines such as Task Arithmetic (69.1%), Ties Merging (72.9%), and LiNeS (74.1%). Notably, even with extremely limited data (1%), `SyMerge` achieves 78.1% accuracy, significantly outperforming the baselines. This result confirms that our expert-guided self-labeling strategy creates a reliable learning signal even from minimal data, making `SyMerge` highly suitable for practical scenarios where data collection is expensive or restricted.

## C   PROOF FOR PROPOSITION 1

We provide a proof for Proposition 1 in the main paper.

**Proposition 1.** *Assume cross-task linearity (Zhou et al., 2024) so that $f(x; \frac{1}{2}(\theta_i + \theta_j)) \approx \frac{1}{2}f(x; \theta_i) + \frac{1}{2}f(x; \theta_j)$, and suppose the loss function is convex in its output. Then the merged model $f_{merge}(x) = f(x; \frac{1}{2}(\theta_i + \theta_j))$ has an expected loss below the case of weight disentanglement.*

*Proof.* Let $f_i(x) \triangleq f(x; \theta_i)$ and $f_j(x) \triangleq f(x; \theta_j)$. The expected loss on task $j$ is denoted by $\mathcal{L}_j(f)$. We assume Cross-Task Linearity (CTL), so $f(x; \frac{1}{2}(\theta_i + \theta_j)) \approx \frac{1}{2}f_i(x) + \frac{1}{2}f_j(x)$. Given that the loss function $L_j$ is convex with respect to its output, Jensen's inequality provides a general upper bound for the merged model's loss:

$$\mathcal{L}_j(f_{\text{merge}}) \leq \frac{1}{2}\mathcal{L}_j(f_i) + \frac{1}{2}\mathcal{L}_j(f_j). \tag{1}$$

We now analyze this bound under two conditions for $\theta_i = \theta_0 + \tau_i$, where $\theta_0$ is a pre-trained model and $\tau_i$ is the task vector for task $i$.

**Case A: Weight Disentanglement.** This condition assumes that the task vector $\tau_i$ does not affect performance on task $j$, i.e., $\mathcal{L}_j(f_i) = \mathcal{L}_j(f_0)$. Substituting this into equation 1 yields:

$$\mathcal{L}_j(f_{\text{merge}}) \leq \frac{1}{2}\mathcal{L}_j(f_0) + \frac{1}{2}\mathcal{L}_j(f_j). \tag{2}$$

**Case B: Synergistic Effect.** This condition assumes that $\tau_i$ improves performance on task $j$, implying $\mathcal{L}_j(f_i) < \mathcal{L}_j(f_0)$. We can write this as $\mathcal{L}_j(f_i) = \mathcal{L}_j(f_0) - \epsilon_{ij}$ for some $\epsilon_{ij} > 0$. This gives a tighter bound:

$$\mathcal{L}_j(f_{\text{merge}}) \leq \frac{1}{2}(\mathcal{L}_j(f_0) - \epsilon_{ij}) + \frac{1}{2}\mathcal{L}_j(f_j) = \left(\frac{1}{2}\mathcal{L}_j(f_0) + \frac{1}{2}\mathcal{L}_j(f_j)\right) - \frac{\epsilon_{ij}}{2}. \tag{3}$$

Comparing equation 2 and equation 3, the upper bound on the loss is strictly lower by $\frac{\epsilon_{ij}}{2}$ under the synergistic effect. This proves that merging synergistic task vectors yields a superior model. □

Table C: **Performance details of merged ViT-B/32 models** when training different layers with merging coefficients for task-specific adjustments. We observe that consistently high performance can be achieved regardless of which task-specific single layer is trained.

| Layer | SUN | Cars | RES. | Euro | SVH. | GTS. | MNI. | DTD | Avg. |
|---|---|---|---|---|---|---|---|---|---|
| 1 | 73.7 | 78.0 | 95.4 | 99.9 | 96.3 | 98.7 | 99.5 | 79.1 | 90.1 |
| 2 | 74.3 | 78.2 | 95.8 | 99.9 | 96.1 | 98.7 | 99.5 | 79.0 | 90.2 |
| 3 | 74.4 | 78.5 | 95.9 | 99.9 | 96.5 | 98.8 | 99.5 | 79.3 | 90.4 |
| 4 | 74.5 | 78.4 | 95.9 | 99.8 | 96.7 | 98.7 | 99.6 | 79.2 | 90.4 |
| 5 | 74.6 | 78.4 | 96.0 | 99.9 | 96.5 | 99.0 | 99.6 | 79.1 | 90.4 |
| 6 | 74.6 | 78.4 | 95.2 | 99.8 | 96.2 | 98.9 | 99.1 | 79.5 | 90.2 |
| 7 | 74.7 | 78.1 | 95.6 | 99.3 | 96.4 | 98.7 | 99.3 | 79.8 | 90.2 |
| 8 | 74.8 | 77.9 | 95.9 | 99.6 | 96.2 | 98.1 | 99.1 | 79.5 | 90.1 |
| 9 | 75.0 | 78.7 | 95.8 | 99.9 | 96.4 | 98.6 | 99.5 | 79.5 | 90.4 |
| 10 | 74.8 | 78.5 | 95.8 | 99.9 | 96.2 | 98.9 | 99.5 | 79.4 | 90.4 |
| 11 | 75.2 | 78.6 | 95.9 | 99.9 | 96.5 | 98.6 | 99.3 | 79.7 | 90.5 |
| 12 | 75.0 | 77.8 | 95.4 | 99.7 | 95.8 | 97.5 | 99.3 | 79.5 | 90.0 |
| Classifier | 74.3 | 79.3 | 94.8 | 99.0 | 95.7 | 98.5 | 99.2 | 80.2 | 90.1 |
| Late | 71.2 | 70.8 | 94.3 | 99.2 | 96.1 | 98.8 | 99.3 | 77.2 | 88.4 |
| Early | 71.7 | 77.2 | 95.1 | 99.9 | 95.6 | 98.7 | 97.7 | 79.5 | 89.4 |
| All | 68.4 | 72.1 | 92.7 | 93.9 | 87.7 | 98.5 | 98.3 | 78.0 | 86.2 |

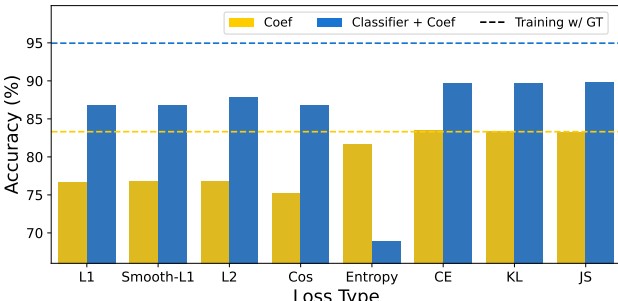

Figure C: **Comparison of training with various loss functions.** The yellow bars represent training only the coefficients, while the blue bars indicate the joint training of the classifier and coefficients. The dashed line corresponds to training with ground truth labels using cross-entropy loss, which is the upper bound.

## D    More Empirical Analyses

### D.1    Detailed Results on Task-specific Layers

Our proposed method, `SyMerge`, introduces the concept of incorporating a task-specific layer to balance shared and task-specific representations during model merging. This approach allows effective adaptation while maintaining efficiency to address task conflicts in multi-task learning. ViT-B/32 is employed and initialized using Task Arithmetic-merged weights across all layers. When a particular layer is marked as task-specific, it is split into unique versions for each task, resulting in eight task-specific layers. While these layers are trained exclusively on corresponding task data, the remaining layers update only the merging coefficients using data from all tasks. Table C presents detailed task-level results. 'Early' refer to layers 1 to 6-th, while 'Late' refer to layers 7 to 12-th. Training both the merging coefficients and task-specific layers achieves performance comparable to or slightly better than training merging coefficients alongside the classifier. These findings highlight the minimal impact of task-specific layer choice on overall performance, reaffirming the robustness of our approach. The results also show consistent average performance across layers, demonstrating that merging coefficients effectively balance representations across tasks, mitigating biases even when other layers remain frozen.

### D.2    On Loss Functions

In Figure C, we analyze the impact of various loss functions on aligning predictions between individual models and the merged model. When training only the coefficients, distance-based losses (*e.g.*, L1, Smooth-L1, L2, and Cosine) demonstrate relatively lower performance compared to distribution-based losses (*e.g.*, KL-divergence, JS-divergence, cross-entropy, and entropy). In

Table D: **Spearman correlation of losses with ground truth cross-entropy loss** for (a) ViT-B/32, (b) ViT-B/16, and (c) ViT-L/14. Values closer to 1 indicate that the corresponding loss function exhibits better alignment with the loss computed using the ground truth.

| | | SUN397 | Cars | RESISC45 | EuroSAT | SVHN | GTSRB | MNIST | DTD |
|---|---|---|---|---|---|---|---|---|---|
| Entropy | Before | 0.66 | 0.72 | 0.80 | 0.86 | 0.92 | 0.92 | 1.00 | 0.56 |
| | After | 0.32 | -0.47 | 0.79 | 0.94 | 0.83 | 0.41 | 0.96 | 0.47 |
| | Δ | (-0.34) | (-1.19) | (-0.01) | (+0.08) | (-0.09) | (-0.51) | (-0.04) | (-0.09) |
| Ours | Before | 0.80 | 0.86 | 0.97 | 1.00 | 0.98 | 0.99 | 1.00 | 0.84 |
| | After | 0.82 | 0.88 | 0.99 | 1.00 | 0.99 | 1.00 | 1.00 | 0.88 |
| | Δ | (+0.02) | (+0.02) | (+0.02) | (+0.00) | (+0.01) | (+0.01) | (+0.00) | (+0.04) |

(a) ViT-B/32

| | | SUN397 | Cars | RESISC45 | EuroSAT | SVHN | GTSRB | MNIST | DTD |
|---|---|---|---|---|---|---|---|---|---|
| Entropy | Before | 0.85 | 0.93 | 0.95 | 0.98 | 0.96 | 0.96 | 1.00 | 0.79 |
| | After | 0.63 | 0.76 | 0.94 | 0.81 | 0.79 | 0.95 | 0.82 | 0.83 |
| | Δ | (-0.22) | (-0.17) | (-0.01) | (-0.17) | (-0.17) | (-0.01) | (-0.18) | (+0.04) |
| Ours | Before | 0.87 | 0.96 | 0.98 | 1.00 | 0.98 | 1.00 | 1.00 | 0.89 |
| | After | 0.89 | 0.97 | 1.00 | 1.00 | 0.99 | 1.00 | 1.00 | 0.91 |
| | Δ | (+0.02) | (+0.01) | (+0.02) | (+0.00) | (+0.01) | (+0.00) | (+0.00) | (+0.02) |

(b) ViT-L/14

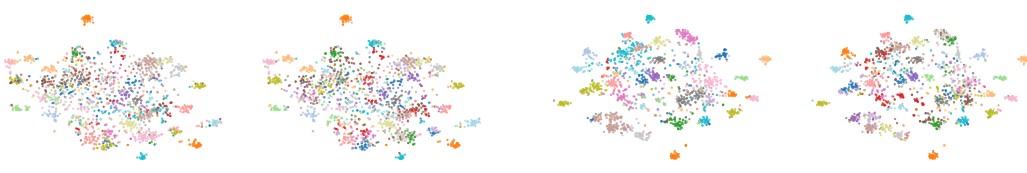

(a) Before training (left: EM Merging   right: GT)          (b) After training (left: EM Merging   right: GT)

Figure D: **t-SNE visualization results on the DTD dataset before (a) and after (b) entropy minimization training.** The plots on the left use merged model predictions, while those on the right use ground truth labels. Training improves clustering, but prediction-ground truth mismatch remains.

contrast, when both the classifier and coefficients are trained jointly, most loss functions achieve strong performance under our method. However, entropy minimization, as explored in previous works (Tang et al., 2024b; Yang et al., 2024b), leads to a significant decline in performance during joint training.

Notably, in Table D, we observe that the entropy loss often fails to provide gradients in the correct direction due to its divergence from the ground truth (GT)-based loss. The table highlights that the cross-entropy loss, in contrast, aligns more closely with the GT, resulting in more accurate gradient directions. This highlights the importance of incorporating label information for guiding classifier training in the merged model.

To further support this point, Figure D shows the t-SNE visualization for the DTD dataset before and after entropy minimization training. While feature clustering improves after training, as shown in (b) compared to (a), a noticeable discrepancy remains between the predictions and the ground truth labels. These observations emphasize the importance of label guidance, even if incomplete, for achieving better alignment with the GT and improving the performance of the merged model.

**Loss correlation.** We conduct further experiments to validate whether the loss correlation shown in Figure 4 of the main paper remains consistent across different backbones (ViT-{B/32, L/14}). Table D shows the loss correlation results for 8 vision tasks using the merged ViT-B/32 and ViT-L/14 models. We observe a significant drop in correlation after training with entropy minimization. In contrast, the cross-entropy loss employed in our self-labeling approach maintains consistently

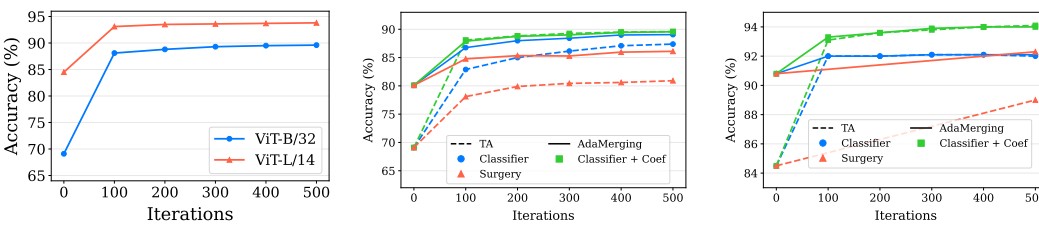

(a) Learning curves across models.    (b) Impact of initialization on ViT-B/32 and ViT-L/14.

Figure E: **Learning curves and analysis.** (a) Comparison of average accuracy for ViT-B/32 and ViT-L/14 across training iterations. (b) Impact of initializations on ViT-B/32 (left) and ViT-L/14 (right), comparing fixed-value initialization from Task Arithmetic (dashed lines) and learned-value one in AdaMerging (solid lines).

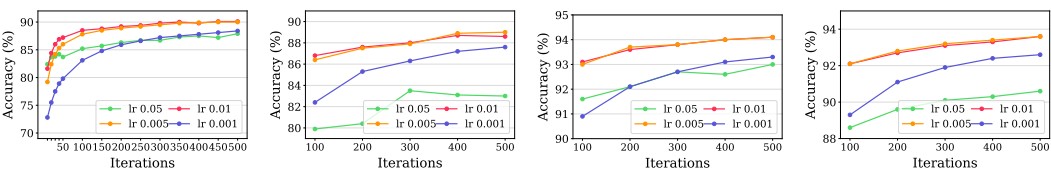

(a) 8 tasks on ViT-B/32. (b) 20 tasks on ViT-B/32. (c) 8 tasks on ViT-L/14. (d) 20 tasks on ViT-L/14.

Figure F: **Impact of learning rate on performance.** Average accuracy curves across training iterations for ViT-B/32 and ViT-L/14 under varying learning rates. We report results for both 8-task and 20-task settings to analyze the hyperparameter sensitivity.

high correlation across tasks. These findings are consistent with the results reported for ViT-B/32 in the main paper, suggesting that our self-labeling approach with the cross-entropy loss could be more effective than the entropy minimization loss.

### D.3 CONVERGENCE ANALYSIS

Figure E illustrates the learning curve of `SyMerge` applied to the 8 vision tasks over training iterations. In Figure Ea, we compare the performance of ViT-B/32 and ViT-L/14 models. The results demonstrate that our method achieves near-optimal performance within the first 100 iterations and converges quickly, regardless of model size. Notably, larger models like ViT-L/14 consistently achieve higher final accuracies, followed by ViT-B/32, which highlights the advantages of model capacity in capturing task-specific knowledge more effectively. Figure Eb shows that when training a task-specific adapter only, the Surgery model is sensitive to initialization. In contrast, our method jointly optimizes the merging coefficient and task-specific layer, making it robust to initialization sensitivity. Figure F shows the convergence of ViT-B/32 and ViT-L/14 under various learning rates for 8-task and 20-task settings. Excluding the extreme learning rate case, our method exhibits rapid convergence and robust performance across most settings. These observations demonstrate the robustness and efficiency of `SyMerge` across different model scales and hyperparameter settings.

### D.4 EMPIRICAL VERIFICATION OF CROSS-TASK LINEARITY

A key theoretical assumption in our work is Cross-Task Linearity (CTL) (Zhou et al., 2024), which posits that the representation of a merged model can be approximated by the linear interpolation of individual model representations. To rigorously validate this assumption, we conducted a quantitative analysis comparing the two interpolation schemes using ViT-B/32.

Specifically, we define the parameter-space interpolation as $\theta(\alpha) = (1 - \alpha)\theta_{\text{src}} + \alpha\theta_{\text{tgt}}$ and the representation-space interpolation as $F_{\text{rep}}(\alpha) = (1-\alpha)f(x; \theta_{\text{src}}) + \alpha f(x; \theta_{\text{tgt}})$. If CTL holds strictly, the output of the merged model $f(x; \theta(\alpha))$ should align closely with $F_{\text{rep}}(\alpha)$. We selected SUN397 as the source task ($\theta_{\text{src}}$) and paired it with seven other target tasks ($\theta_{\text{tgt}}$) to measure the Cosine Similarity between $f(x; \theta(\alpha))$ and $F_{\text{rep}}(\alpha)$ across varying interpolation coefficients $\alpha \in [0.1, 0.9]$.

Table E presents the results. We observe that the cosine similarity remains remarkably high across all task pairs and $\alpha$ values. Most notably, even at the interpolation midpoint ($\alpha = 0.5$), the similarity scores remain consistently high. These findings empirically demonstrate that within the shared

Table E: **Empirical Verification of Cross-Task Linearity (CTL).** We measure the cosine similarity between the output of a parameter-interpolated model $f(x; \theta(\alpha))$ and the output interpolation $(1 - \alpha)f(x; \theta_{\text{src}}) + \alpha f(x; \theta_{\text{tgt}})$ on ViT-B/32. Using SUN397 as the source task ($\theta_{\text{src}}$), the consistently high similarity scores across all coefficients $\alpha$ (even at $\alpha = 0.5$) confirm that the CTL assumption holds robustly in our experimental setting.

| Task | Interpolation Coefficient ($\alpha$) | | | | | | | | |
|---|---|---|---|---|---|---|---|---|---|
| | 0.1 | 0.2 | 0.3 | 0.4 | 0.5 | 0.6 | 0.7 | 0.8 | 0.9 |
| Cars | 1.00 | 0.99 | 0.98 | 0.97 | 0.97 | 0.97 | 0.98 | 0.99 | 1.00 |
| RESISC45 | 0.99 | 0.98 | 0.97 | 0.96 | 0.95 | 0.95 | 0.96 | 0.98 | 0.99 |
| EuroSAT | 0.99 | 0.98 | 0.96 | 0.95 | 0.94 | 0.94 | 0.95 | 0.97 | 0.99 |
| SVHN | 0.99 | 0.97 | 0.94 | 0.90 | 0.87 | 0.85 | 0.87 | 0.92 | 0.97 |
| GTSRB | 0.99 | 0.96 | 0.93 | 0.90 | 0.88 | 0.88 | 0.90 | 0.94 | 0.98 |
| MNIST | 0.99 | 0.97 | 0.94 | 0.92 | 0.90 | 0.89 | 0.91 | 0.95 | 0.98 |
| DTD | 0.99 | 0.98 | 0.96 | 0.96 | 0.96 | 0.96 | 0.97 | 0.99 | 1.00 |

Table F: Ablation study on confidence-based filtering. We report the mean $\pm$ standard deviation over 5 runs. Applying the filtering consistently improves performance across different numbers of tasks and model scales.

| Setting | Model | Accuracy (%) | |
|---|---|---|---|
| | | w/o Filtering | w/ Filtering |
| 8 tasks | ViT-B/32 | $89.7 \pm 0.1$ | $90.1 \pm 0.1$ (+0.4) |
| 14 tasks | ViT-B/32 | $88.3 \pm 0.2$ | $88.7 \pm 0.1$ (+0.4) |
| 20 tasks | ViT-B/32 | $88.5 \pm 0.3$ | $88.6 \pm 0.4$ (+0.1) |
| 8 tasks | ViT-L/14 | $93.9 \pm 0.0$ | $94.1 \pm 0.0$ (+0.2) |

solution basin derived from a common pre-trained model, the encoder's behavior exhibits strong linearity. Consequently, this confirms that the CTL assumption serves as a robust approximation in our experimental regime, supporting the validity of our theoretical proposition.

### D.5 IMPACT OF CONFIDENCE-BASED FILTERING

To verify the effect of confidence-based filtering, we evaluate the model without it. As detailed in Table F, employing the filtering strategy yields consistent but modest performance gains across varying task counts and model scales (e.g., +0.4%p on 8 tasks). Crucially, even without filtering, **SyMerge** maintains state-of-the-art performance (e.g., 89.7% on 8 tasks) significantly outperforming existing baselines. This confirms that while filtering serves as an effective safeguard against spurious supervision from potentially noisy experts, the core efficacy of **SyMerge** stems from the synergistic joint optimization framework itself.

### D.6 COMPUTATIONAL EFFICIENCY ANALYSIS

To evaluate the resource efficiency of **SyMerge**, we measured the peak GPU memory usage and the number of trainable parameters across 8 vision tasks. For a fair comparison, we standardized all methods to employ sequential updates (task-by-task optimization) with a batch size of 16. This adaptation was necessary because the original simultaneous update schemes of baselines like AdaMerging and WeMoE incur excessive memory costs, often leading to Out-Of-Memory (OOM) errors on large backbones (e.g., ViT-L/14).

As shown in Table G, **SyMerge** demonstrates remarkable efficiency. Its peak memory usage is comparable to AdaMerging, the most lightweight baseline, and significantly lower than Surgery and WEMoE. Furthermore, **SyMerge** introduces negligible trainable parameters (e.g., <0.6M for ViT-L/14). This confirms that our method achieves state-of-the-art performance with a minimal computational footprint, making it practical for resource-constrained environments.

Table G: **Computational efficiency.** We report the peak GPU memory usage and the number of trainable parameters during adaptation on 8 vision tasks. To prevent OOM errors on baselines, all methods are evaluated under a standardized sequential update setting (batch size=16). **SyMerge** achieves high efficiency, comparable to the most lightweight baseline.

| Method | Peak GPU Mem (GB) | | Params (M) | |
|---|---|---|---|---|
| | ViT-B/32 | ViT-L/14 | ViT-B/32 | ViT-L/14 |
| AdaMerging (Yang et al., 2024b) | 7.09 | 22.88 | 0.00 | 0.00 |
| Surgery (Yang et al., 2024a) | 7.13 | 30.44 | 0.13 | 0.20 |
| WEMoE (Tang et al., 2024b) | 8.01 | 33.57 | 7.16 | 25.39 |
| **SyMerge** | 7.12 | 23.71 | 0.39 | 0.59 |

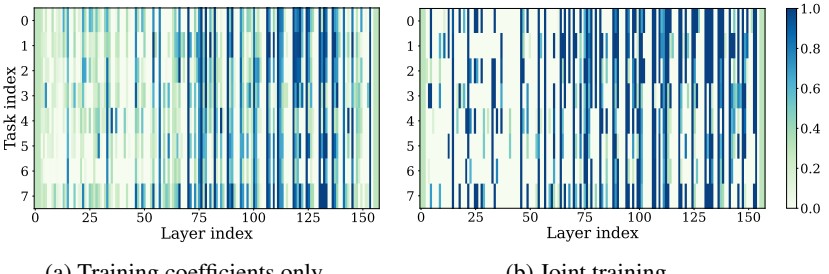

(a) Training coefficients only      (b) Joint training

Figure G: Impact of joint training on merging coefficient sparsity on ViT-B/32.

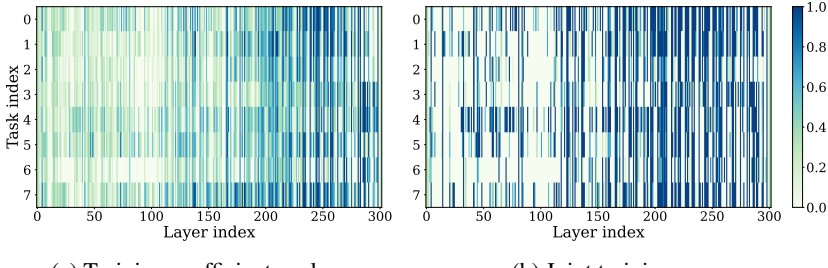

(a) Training coefficients only      (b) Joint training

Figure H: Impact of joint training on merging coefficient sparsity on ViT-L/14.

### D.7 SPARSITY VISUALIZATION

Inspired by the literature (Yu et al., 2024; Yadav et al., 2024; Davari & Belilovsky, 2024), which highlights how redundant parameters degrade performance due to conflicts among task-specific parameters, we investigate this phenomenon in our method. We analyze the learned merging coefficients to observe their impact, specifically exploring *sparsity* after training them alone and jointly with the classifier.

To confirm this trend across various backbones, Figure G and H provide the layer-wise merging coefficients for ViT-B/32 and ViT-L/14. All models are optimized using the cross-entropy loss with the predictions of individual models as guidance. In these figures, (a) represents training only the coefficients, while (b) includes joint training of the classifier and coefficients.

We find that jointly training a task-specific layer, like the classifier, with merging coefficients increases the proportion of coefficients concentrated near zero (*i.e.*, smaller than 1e-5), leading to improved accuracy. For example, joint training boosts the share of near-zero coefficients from 37.2% to 55.9%, with a corresponding accuracy improvement from 84.6% to 90.1%. This trend is consistent across backbones like ViT-L/14, where the proportion of sparse coefficients similarly rises (from 23.3% to 55.0%) as does the average accuracy (from 91.5% to 94.1%). These results indicate that training the classifier complements sparsity in the merging coefficients, effectively reducing task conflicts by pruning unnecessary parameters and enhancing performance.

Table H: Ablations of the corrupted test dataset on ViT-B/32.

| Method | Cars | EuroSAT | RESISC45 | GTSRB | Avg. | Cars | EuroSAT | RESISC45 | GTSRB | Avg. |
|---|---|---|---|---|---|---|---|---|---|---|
| | | | Original Test Set | | | | Corrupted Test Set (Motion Blur) | | | |
| Pretrained | 59.6 | 45.0 | 60.2 | 32.6 | 49.4 | 57.4 | 23.8 | 55.5 | 22.6 | 39.8 |
| Individual | 77.7 | 99.9 | 96.1 | 98.7 | 93.1 | 75.1 | 75.2 | 94.1 | 95.3 | 84.9 |
| Task Arithmetic | 67.0 | 94.0 | 82.6 | 75.1 | 79.7 | 63.8 | 55.1 | 78.0 | 53.0 | 62.5 |
| Ties-Merging | 67.5 | 83.7 | 79.3 | 65.4 | 74.0 | 65.6 | 52.3 | 74.0 | 44.9 | 59.2 |
| PCB-Merging | 70.8 | 89.6 | 84.6 | 80.7 | 81.4 | 68.4 | 57.0 | 79.5 | 60.2 | 66.3 |
| LiNeS | 69.5 | 96.3 | 84.6 | 82.4 | 83.2 | 67.0 | 61.1 | 80.4 | 61.3 | 67.5 |
| Consensus TA | 67.1 | 95.4 | 78.0 | 84.9 | 81.4 | 63.9 | 59.5 | 56.2 | 80.3 | 65.0 |
| Iso-CTS | 73.6 | 96.3 | 91.3 | 93.4 | 88.6 | 69.7 | 67.9 | 76.4 | 87.8 | 75.5 |
| AdaMerging | 76.2 | 96.0 | 87.5 | 97.0 | 89.2 | 71.7 | 69.4 | 83.3 | 91.9 | 79.1 |
| Surgery w/ Ada | 73.4 | 98.5 | 91.2 | 97.8 | 90.2 | 69.9 | 73.7 | 88.7 | 90.1 | 80.6 |
| WEMoE | 79.1 | 99.3 | 95.5 | 99.1 | 93.2 | 76.6 | 54.5 | 93.4 | 96.1 | 80.2 |
| **SyMerge** | 77.6 | 99.4 | 95.3 | 98.3 | 92.7 | 74.7 | 76.0 | 93.2 | 93.6 | 84.4 |
| | | | Corrupted Test Set (Impulse Noise) | | | | Corrupted Test Set (Gaussian Noise) | | | |
| Pretrained | 49.9 | 11.5 | 38.1 | 12.4 | 28.0 | 54.4 | 6.4 | 48.8 | 14.2 | 31.0 |
| Individual | 69.1 | 14.3 | 79.0 | 66.9 | 57.3 | 73.9 | 14.0 | 90.1 | 74.6 | 63.2 |
| Task Arithmetic | 58.2 | 16.3 | 56.9 | 24.8 | 39.0 | 62.8 | 26.3 | 68.5 | 36.5 | 48.5 |
| Ties-Merging | 59.7 | 13.7 | 55.0 | 20.9 | 37.3 | 63.6 | 23.3 | 66.6 | 29.3 | 45.7 |
| PCB-Merging | 62.3 | 15.7 | 61.0 | 28.4 | 41.8 | 66.5 | 26.7 | 73.0 | 41.4 | 51.9 |
| LiNeS | 60.6 | 18.5 | 58.3 | 28.1 | 41.4 | 65.7 | 28.9 | 70.2 | 40.7 | 51.4 |
| Consensus TA | 58.2 | 17.9 | 24.4 | 56.9 | 39.4 | 62.9 | 29.0 | 36.8 | 69.4 | 49.5 |
| Iso-CTS | 62.9 | 20.4 | 33.5 | 64.2 | 45.2 | 69.0 | 33.5 | 46.5 | 77.1 | 56.5 |
| AdaMerging | 62.7 | 14.3 | 65.5 | 53.8 | 49.0 | 69.6 | 17.4 | 73.9 | 63.1 | 56.0 |
| Surgery w/ Ada | 62.7 | 12.4 | 68.3 | 48.8 | 48.1 | 68.6 | 11.9 | 82.3 | 59.7 | 55.6 |
| WEMoE | 70.3 | 11.6 | 83.5 | 9.6 | 43.7 | 74.9 | 11.6 | 90.6 | 8.0 | 46.2 |
| **SyMerge** | 69.0 | 13.1 | 79.5 | 63.6 | 56.3 | 74.4 | 12.6 | 88.8 | 72.4 | 62.1 |
| | | | Corrupted Test Set (Pixelate) | | | | Corrupted Test Set (Spatter) | | | |
| Pretrained | 58.5 | 23.8 | 59.3 | 23.9 | 41.4 | 48.5 | 20.7 | 45.8 | 25.6 | 35.1 |
| Individual | 76.7 | 96.7 | 95.6 | 96.0 | 91.3 | 71.1 | 88.5 | 83.6 | 94.8 | 84.5 |
| Task Arithmetic | 65.8 | 78.5 | 81.1 | 55.9 | 70.3 | 57.5 | 55.9 | 67.2 | 56.8 | 59.3 |
| Ties-Merging | 66.8 | 68.7 | 78.9 | 46.3 | 65.2 | 58.2 | 45.5 | 64.6 | 46.5 | 53.7 |
| PCB-Merging | 69.5 | 75.4 | 84.0 | 62.1 | 72.8 | 61.3 | 52.9 | 70.5 | 63.0 | 61.9 |
| LiNeS | 68.4 | 84.7 | 83.3 | 61.7 | 74.5 | 59.8 | 59.6 | 68.8 | 64.8 | 63.3 |
| Consensus TA | 65.9 | 82.6 | 58.5 | 83.5 | 72.6 | 57.0 | 65.4 | 58.4 | 68.6 | 62.4 |
| Iso-CTS | 71.7 | 82.0 | 76.2 | 90.4 | 80.1 | 64.1 | 71.8 | 76.0 | 74.2 | 71.5 |
| AdaMerging | 71.2 | 89.8 | 81.7 | 90.8 | 83.4 | 66.5 | 14.6 | 74.5 | 92.0 | 61.9 |
| Surgery w/ Ada | 71.8 | 94.9 | 90.0 | 89.2 | 86.5 | 63.1 | 87.0 | 74.7 | 90.5 | 78.8 |
| WEMoE | 77.6 | 96.5 | 94.5 | 96.0 | 91.1 | 72.0 | 11.6 | 86.0 | 95.1 | 66.2 |
| **SyMerge** | 76.4 | 96.5 | 95.1 | 95.4 | 90.8 | 71.2 | 87.9 | 82.9 | 94.4 | 84.1 |
| | | | Corrupted Test Set (Contrast) | | | | Corrupted Test Set (JPEG Compression) | | | |
| Pretrained | 51.5 | 15.0 | 43.3 | 23.6 | 33.4 | 58.3 | 22.3 | 58.6 | 24.3 | 40.9 |
| Individual | 68.1 | 22.2 | 56.7 | 82.6 | 57.4 | 77.2 | 67.4 | 95.4 | 92.9 | 83.2 |
| Task Arithmetic | 55.2 | 28.0 | 48.2 | 48.5 | 45.0 | 65.9 | 65.9 | 80.3 | 53.3 | 66.4 |
| Ties-Merging | 57.5 | 34.3 | 52.4 | 42.5 | 46.6 | 66.8 | 51.4 | 78.1 | 43.1 | 59.8 |
| PCB-Merging | 58.6 | 33.7 | 53.3 | 56.4 | 50.5 | 70.3 | 60.7 | 83.5 | 59.1 | 68.4 |
| LiNeS | 58.2 | 29.4 | 49.0 | 55.3 | 48.0 | 68.9 | 68.7 | 82.5 | 59.9 | 70.0 |
| Consensus TA | 55.7 | 31.1 | 47.9 | 49.9 | 46.2 | 66.5 | 65.4 | 53.7 | 82.7 | 67.1 |
| Iso-CTS | 63.0 | 26.6 | 69.6 | 55.0 | 53.5 | 72.5 | 57.8 | 71.1 | 89.8 | 72.8 |
| AdaMerging | 65.8 | 39.7 | 60.2 | 90.7 | 64.1 | 72.1 | 65.4 | 82.9 | 87.3 | 76.9 |
| Surgery w/ Ada | 62.0 | 21.1 | 50.6 | 83.0 | 54.2 | 72.1 | 66.0 | 89.8 | 84.9 | 78.2 |
| WEMoE | 70.9 | 47.0 | 65.7 | 93.1 | 69.2 | 78.0 | 70.5 | 94.3 | 92.8 | 83.9 |
| **SyMerge** | 67.6 | 20.5 | 56.5 | 81.3 | 56.5 | 77.3 | 65.0 | 94.3 | 91.5 | 82.0 |

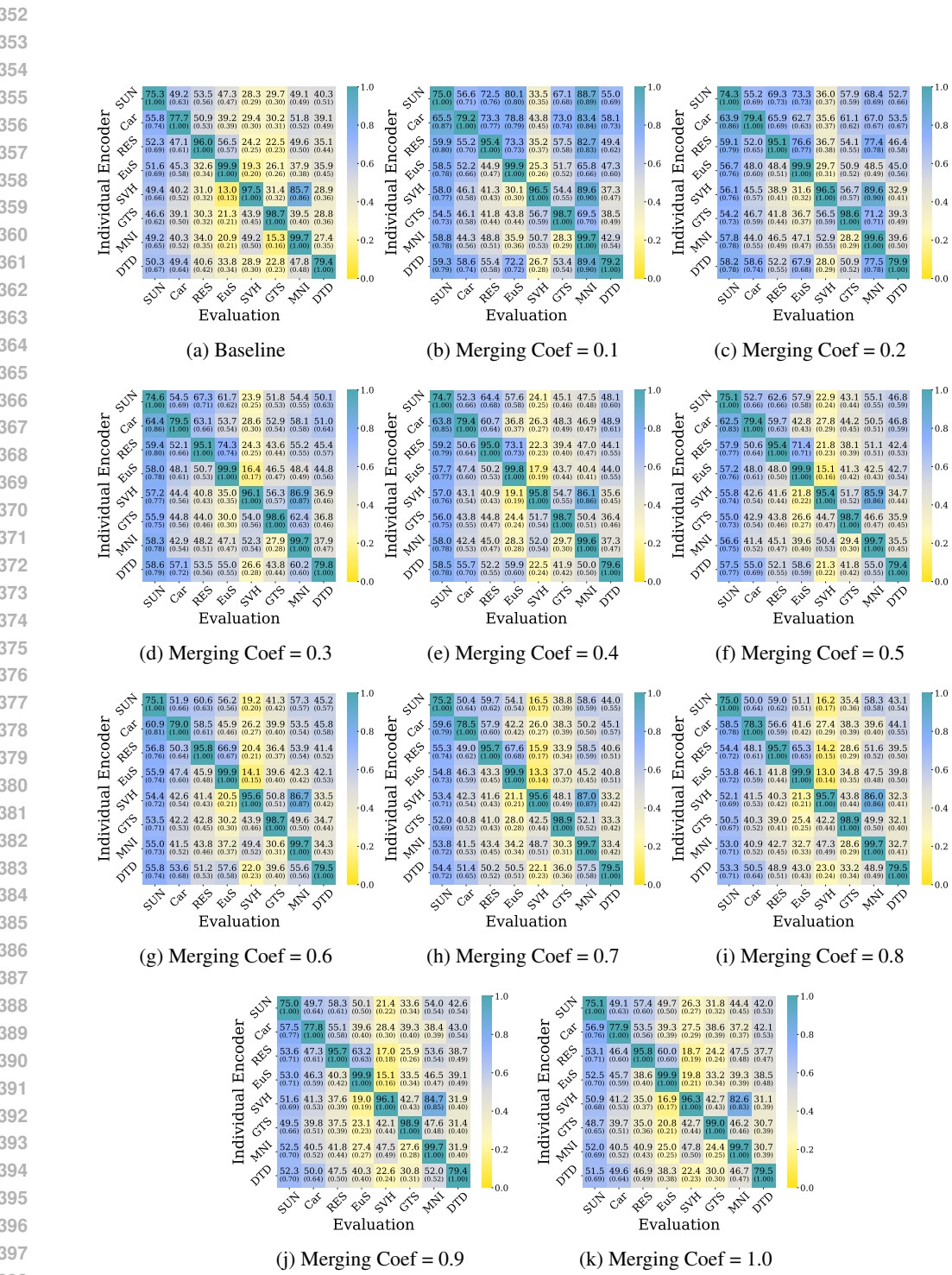

Figure I: Additional results for cross-task evaluations across diverse encoders and heads from different tasks.

Table I: Multi-task performance when merging ViT-B/32 models on 8 tasks

| Method | SUN397 | Cars | RESISC45 | EuroSAT | SVHN | GTSRB | MNIST | DTD | Avg. |
|---|---|---|---|---|---|---|---|---|---|
| Pretrained | 62.3 | 59.7 | 60.7 | 45.5 | 31.4 | 32.6 | 48.5 | 43.8 | 48.0 |
| Individual | 75.3 | 77.7 | 96.1 | 99.7 | 97.5 | 98.7 | 99.7 | 79.4 | 90.5 |
| Task Arithmetic (Ilharco et al., 2023) | 55.2 | 54.9 | 66.7 | 78.9 | 80.2 | 69.7 | 97.3 | 50.4 | 69.1 |
| Ties Merging (Yadav et al., 2024) | 65.0 | 64.4 | 74.8 | 77.4 | 81.2 | 69.3 | 96.5 | 54.5 | 72.9 |
| PCB Merging (Du et al., 2024) | 62.9 | 62.0 | 77.1 | 80.1 | 87.5 | 78.5 | 98.7 | 58.4 | 75.6 |
| LiNeS w/ TA (Wang et al., 2025) | 63.7 | 63.9 | 75.1 | 86.1 | 79.4 | 72.2 | 96.2 | 56.5 | 74.1 |
| Consensus TA (Wang et al., 2024) | 62.9 | 61.0 | 71.0 | 82.7 | 86.8 | 79.3 | 98.1 | 57.2 | 74.9 |
| TSV-M (Gargiulo et al., 2025) | 68.8 | 72.0 | 85.9 | 94.7 | 90.9 | 91.2 | 99.2 | 69.1 | 84.0 |
| ISO-CTS (Marczak et al., 2025a) | 70.3 | 73.7 | 89.1 | 95.0 | 86.3 | 90.9 | 99.0 | 69.4 | 84.2 |
| EMR-Merging (Huang et al., 2024) | 75.2 | 72.8 | 93.5 | 99.5 | 96.9 | 98.1 | 99.6 | 74.4 | 88.7 |
| AdaMerging (Yang et al., 2024b) | 64.5 | 68.1 | 79.2 | 93.8 | 87.0 | 91.9 | 97.5 | 59.1 | 80.1 |
| Surgery w/ Ada. (Yang et al., 2024a) | 71.2 | 72.0 | 92.3 | 99.0 | 92.2 | 97.9 | 99.0 | 76.1 | 87.5 |
| WEMoE (Tang et al., 2024b) | 74.1 | 77.4 | 93.7 | 99.1 | 96.2 | 98.9 | 99.6 | 76.4 | 89.4 |
| ProbSurgery w/ Ada. (Wei et al., 2025) | 70.3 | 71.6 | 91.9 | 98.8 | 92.5 | 97.9 | 99.0 | 77.5 | 87.4 |
| **SyMerge** | 74.7 | 79.0 | 95.1 | 98.9 | 95.7 | 98.3 | 99.1 | 80.0 | 90.1 |

Table J: Multi-task performance when merging ViT-B/32 models on 14 tasks.

| Method | SUN | Cars | RES. | Euro | SVH. | GTS. | MNI. | DTD | C100 | FER | Flower | Pet | PCAM | STL | Avg. |
|---|---|---|---|---|---|---|---|---|---|---|---|---|---|---|---|
| Pretrained | 62.3 | 59.7 | 60.7 | 45.5 | 31.4 | 32.6 | 48.5 | 43.8 | 64.2 | 39.0 | 66.3 | 87.4 | 60.6 | 97.1 | 57.1 |
| Individual | 75.3 | 77.7 | 96.1 | 99.7 | 97.5 | 98.7 | 99.7 | 79.4 | 89.1 | 72.5 | 90.4 | 91.8 | 87.8 | 97.8 | 89.5 |
| Task Arithmetic (Ilharco et al., 2023) | 63.9 | 59.5 | 67.5 | 67.7 | 52.9 | 47.0 | 80.8 | 48.2 | 69.6 | 42.9 | 67.6 | 87.5 | 63.2 | 96.7 | 65.4 |
| Ties Merging (Yadav et al., 2024) | 65.1 | 61.8 | 68.3 | 63.7 | 51.3 | 45.9 | 80.0 | 48.7 | 69.7 | 42.4 | 68.1 | 88.0 | 62.1 | 97.2 | 65.2 |
| PCB Merging (Du et al., 2024) | 52.5 | 40.7 | 62.2 | 71.3 | 60.8 | 61.7 | 94.1 | 50.3 | 51.1 | 44.0 | 57.4 | 82.0 | 75.1 | 90.4 | 63.8 |
| LiNeS w/ TA (Wang et al., 2025) | 64.3 | 59.9 | 69.6 | 75.4 | 58.8 | 54.6 | 85.1 | 51.1 | 70.0 | 45.0 | 69.3 | 88.0 | 64.3 | 96.8 | 68.0 |
| Consensus TA (Wang et al., 2024) | 63.8 | 57.5 | 69.5 | 77.9 | 69.2 | 60.4 | 93.7 | 52.4 | 67.3 | 44.4 | 68.9 | 88.1 | 74.6 | 95.9 | 70.3 |
| TSV-M (Gargiulo et al., 2025) | 66.7 | 62.0 | 81.5 | 93.6 | 84.6 | 84.8 | 98.7 | 65.9 | 71.0 | 63.3 | 76.4 | 91.0 | 84.9 | 97.1 | 80.1 |
| ISO-CTS (Marczak et al., 2025a) | 68.9 | 66.1 | 85.7 | 91.8 | 80.5 | 85.3 | 98.4 | 67.3 | 74.9 | 59.3 | 82.0 | 89.0 | 81.8 | 97.4 | 80.6 |
| EMR-Merging (Huang et al., 2024) | 70.4 | 68.3 | 92.5 | 99.0 | 96.1 | 97.6 | 99.5 | 72.0 | 81.2 | 68.5 | 85.9 | 90.7 | 86.7 | 97.3 | 86.1 |
| AdaMerging (Yang et al., 2024b) | 64.3 | 68.5 | 81.7 | 92.6 | 86.6 | 90.8 | 97.5 | 60.2 | 67.3 | 53.1 | 73.8 | 87.9 | 53.8 | 96.3 | 76.7 |
| Surgery w/ Ada. (Yang et al., 2024a) | 69.1 | 71.5 | 89.5 | 97.8 | 90.2 | 95.3 | 98.6 | 73.6 | 73.4 | 66.3 | 86.0 | 92.2 | 82.3 | 98.1 | 84.6 |
| WEMoE (Tang et al., 2024b) | 74.2 | 78.1 | 94.0 | 98.2 | 95.8 | 98.5 | 99.6 | 75.9 | 83.7 | 32.8 | 84.9 | 91.8 | 51.3 | 97.8 | 83.0 |
| ProbSurgery w/ Ada. (Wei et al., 2025) | 68.9 | 68.7 | 91.1 | 98.6 | 92.2 | 95.0 | 98.3 | 77.2 | 72.3 | 64.4 | 87.1 | 91.8 | 84.6 | 98.4 | 84.9 |
| **SyMerge** | 74.0 | 78.7 | 94.6 | 98.8 | 94.6 | 97.7 | 98.9 | 80.0 | 84.9 | 71.0 | 92.2 | 92.2 | 86.1 | 98.1 | 88.7 |

Table K: Multi-task performance when merging ViT-B/32 models on 20 tasks.

| Method | SUN397 | Cars | RESISC45 | EuroSAT | SVHN | GTSRB | MNIST | DTD | CIFAR100 | FER2013 |
|---|---|---|---|---|---|---|---|---|---|---|
| Pretrained | 66.8 | 77.7 | 71.0 | 59.9 | 58.4 | 50.5 | 76.3 | 55.3 | 75.8 | 38.2 |
| Individual | 75.3 | 77.7 | 96.1 | 99.7 | 97.5 | 98.7 | 99.7 | 79.4 | 89.1 | 72.5 |
| Task arithmetic (Ilharco et al., 2023) | 61.8 | 53.0 | 61.9 | 57.5 | 49.8 | 44.6 | 77.9 | 45.6 | 65.4 | 41.4 |
| Ties Merging (Yadav et al., 2024) | 64.6 | 58.7 | 66.4 | 59.7 | 54.9 | 46.7 | 80.1 | 47.5 | 69.0 | 41.8 |
| PCB Merging (Du et al., 2024) | 41.3 | 21.5 | 44.6 | 47.4 | 52.4 | 41.9 | 86.9 | 39.7 | 40.7 | 38.5 |
| LiNeS w/ TA (Wang et al., 2025) | 63.6 | 55.3 | 66.3 | 65.0 | 56.7 | 51.3 | 81.6 | 48.8 | 68.3 | 44.1 |
| Consensus TA (Wang et al., 2024) | 63.7 | 53.4 | 66.3 | 63.8 | 63.5 | 52.2 | 89.5 | 49.4 | 66.3 | 41.1 |
| TSV-M (Gargiulo et al., 2025) | 65.8 | 54.1 | 77.8 | 89.8 | 76.9 | 74.9 | 94.1 | 60.6 | 69.5 | 58.3 |
| ISO-CTS (Marczak et al., 2025a) | 67.0 | 56.2 | 80.4 | 83.2 | 75.4 | 76.5 | 96.4 | 63.9 | 74.0 | 57.0 |
| EMR-Merging (Huang et al., 2024) | 71.0 | 67.6 | 91.1 | 98.6 | 94.4 | 96.7 | 99.4 | 71.0 | 81.1 | 65.7 |
| AdaMerging (Yang et al., 2024b) | 63.7 | 65.5 | 77.9 | 90.8 | 75.0 | 89.3 | 96.7 | 56.2 | 67.7 | 48.0 |
| Surgery w/ Ada. (Yang et al., 2024a) | 67.9 | 69.2 | 89.0 | 97.8 | 86.1 | 95.3 | 98.4 | 71.8 | 73.8 | 63.5 |
| WEMoE (Tang et al., 2024b) | 80.1 | 91.9 | 95.5 | 98.7 | 96.5 | 98.6 | 98.9 | 76.4 | 88.7 | 17.2 |
| ProbSurgery w/ Ada. (Wei et al., 2025) | 67.9 | 64.6 | 91.2 | 98.0 | 90.9 | 95.1 | 98.5 | 74.5 | 67.1 | 60.4 |
| **SyMerge** | 73.4 | 78.2 | 93.7 | 98.0 | 92.4 | 97.1 | 98.8 | 79.8 | 83.7 | 69.5 |

| Method | Flowers | Pet | PCAM | STL10 | CIFAR10 | EMNIST | FMNIST | Food101 | KMNIST | R-SST2 |
|---|---|---|---|---|---|---|---|---|---|---|
| Pretrained | 79.2 | 93.4 | 51.2 | 99.4 | 95.6 | 15.6 | 66.9 | 92.3 | 10.4 | 68.9 |
| Individual | 90.4 | 91.8 | 87.8 | 97.8 | 97.9 | 99.7 | 95.4 | 89.1 | 98.4 | 74.8 |
| Task arithmetic | 63.1 | 86.0 | 65.8 | 94.4 | 91.5 | 39.6 | 73.9 | 72.1 | 12.2 | 57.8 |
| Ties Merging | 66.4 | 87.4 | 63.8 | 96.4 | 92.7 | 41.2 | 73.2 | 79.5 | 12.5 | 60.4 |
| PCB Merging | 43.1 | 71.7 | 68.4 | 85.9 | 80.3 | 62.1 | 74.1 | 34.5 | 27.0 | 52.4 |
| LiNeS w/ TA | 67.0 | 87.4 | 64.7 | 96.0 | 92.5 | 45.5 | 75.4 | 78.0 | 13.8 | 53.2 |
| Consensus TA | 66.6 | 86.3 | 68.9 | 95.8 | 92.5 | 51.9 | 74.5 | 75.5 | 17.0 | 62.4 |
| TSV-M | 72.7 | 90.1 | 83.6 | 96.8 | 94.2 | 94.4 | 84.0 | 79.3 | 44.6 | 70.4 |
| ISO-CTS | 77.6 | 89.5 | 82.9 | 97.0 | 95.1 | 83.9 | 85.7 | 77.4 | 49.6 | 70.3 |
| EMR-Merging | 83.8 | 91.3 | 85.9 | 97.7 | 96.8 | 99.6 | 93.2 | 83.7 | 91.4 | 71.3 |
| AdaMerging | 68.0 | 87.6 | 54.1 | 96.5 | 91.3 | 30.8 | 80.9 | 79.7 | 12.6 | 60.2 |
| Surgery w/ Ada. | 83.6 | 91.7 | 83.7 | 98.2 | 94.6 | 97.0 | 88.9 | 83.9 | 82.1 | 73.8 |
| WEMoE | 98.3 | 96.0 | 51.2 | 99.4 | 98.0 | 99.1 | 37.4 | 94.8 | 10.0 | 49.9 |
| ProbSurgery w/ Ada. | 84.1 | 91.6 | 84.4 | 98.1 | 92.6 | 98.4 | 85.1 | 81.5 | 92.8 | 73.6 |
| **SyMerge** | 91.7 | 92.2 | 85.6 | 97.8 | 95.5 | 98.2 | 90.5 | 85.3 | 95.4 | 76.1 |

Table L: Multi-task performance when merging ViT-L/14 models on 8 tasks.

| Method | SUN397 | Cars | RESISC45 | EuroSAT | SVHN | GTSRB | MNIST | DTD | Avg. |
|---|---|---|---|---|---|---|---|---|---|
| Pretrained | 66.8 | 77.7 | 71 | 59.9 | 58.4 | 50.5 | 76.3 | 55.3 | 64.5 |
| Individual | 82.3 | 92.4 | 97.4 | 100 | 98.1 | 99.2 | 99.7 | 84.1 | 94.2 |
| Task Arithmetic (Ilharco et al., 2023) | 73.9 | 82.1 | 86.6 | 94.1 | 87.9 | 86.7 | 98.9 | 65.6 | 84.5 |
| Ties Merging (Yadav et al., 2024) | 76.5 | 85.0 | 89.3 | 95.7 | 90.3 | 83.3 | 99.0 | 68.8 | 86.0 |
| PCB Merging (Du et al., 2024) | 75.9 | 85.9 | 89.6 | 95.7 | 89.9 | 92.3 | 99.2 | 71.4 | 87.5 |
| LiNeS w/ TA (Wang et al., 2025) | 74.5 | 85.4 | 88.8 | 95.4 | 90.8 | 90.8 | 99.3 | 70.4 | 86.9 |
| Consensus TA (Wang et al., 2024) | 74.9 | 83.0 | 88.2 | 95.4 | 91.3 | 91.5 | 99.1 | 69.6 | 86.6 |
| TSV-M (Gargiulo et al., 2025) | 78.4 | 89.9 | 93.7 | 98.7 | 95.6 | 96.5 | 99.5 | 79.9 | 91.5 |
| ISO-CTS (Marczak et al., 2025a) | 80.6 | 91.7 | 96.0 | 99.2 | 95.1 | 98.5 | 99.5 | 83.0 | 93.0 |
| EMR-Merging (Huang et al., 2024) | 83.2 | 90.7 | 96.8 | 99.7 | 97.9 | 99.1 | 99.7 | 82.7 | 93.7 |
| AdaMerging (Yang et al., 2024b) | 79.0 | 90.3 | 90.8 | 96.2 | 93.4 | 98.0 | 99.0 | 79.9 | 90.8 |
| Surgery w/ Ada. (Yang et al., 2024a) | 80.3 | 90.8 | 94.3 | 98.2 | 94.1 | 98.7 | 99.2 | 82.5 | 92.3 |
| WeMoE (Tang et al., 2024b) | 81.4 | 92.6 | 95.4 | 99.4 | 97.7 | 99.3 | 99.7 | 83.7 | 93.6 |
| ProbSurgery w/ Ada. (Wei et al., 2025) | 79.1 | 91.1 | 95.5 | 99.1 | 95.2 | 99.0 | 99.3 | 83.4 | 92.7 |
| **SyMerge** | 81.8 | 92.6 | 97.2 | 99.7 | 97.8 | 99.1 | 99.5 | 84.8 | 94.1 |

Table M: Multi-task performance when merging ViT-L/14 models on 14 tasks.

| Method | SUN | Cars | RES. | Euro | SVH. | GTS. | MNI. | DTD | C100 | FER | Flower | Pet | PCAM | STL | Avg. |
|---|---|---|---|---|---|---|---|---|---|---|---|---|---|---|---|
| Pretrained | 66.8 | 77.7 | 71.0 | 59.9 | 58.4 | 50.5 | 76.3 | 55.3 | 75.8 | 38.2 | 79.2 | 93.4 | 51.2 | 99.4 | 68.1 |
| Individual | 82.3 | 92.4 | 97.4 | 100 | 98.1 | 99.2 | 99.7 | 84.1 | 93.3 | 76.5 | 97.9 | 95.2 | 90.0 | 99.3 | 93.2 |
| Task Arithmetic (Ilharco et al., 2023) | 71.6 | 79.0 | 80.6 | 86.5 | 75.6 | 65.8 | 96.0 | 60.9 | 83.9 | 43.5 | 80.2 | 94.9 | 79.7 | 99.4 | 78.4 |
| Ties Merging (Yadav et al., 2024) | 73.3 | 78.1 | 84.3 | 87.6 | 84.4 | 79.7 | 98.5 | 63.4 | 81.9 | 47.5 | 78.2 | 94.7 | 76.9 | 99.2 | 80.5 |
| PCB Merging (Du et al., 2024) | 72.2 | 73.3 | 82.4 | 89.3 | 86.2 | 85.2 | 98.9 | 63.6 | 76.8 | 47.7 | 87.4 | 94.4 | 82.4 | 98.7 | 81.3 |
| LiNeS w/ TA (Wang et al., 2025) | 72.6 | 79.7 | 83.1 | 89.9 | 78.2 | 72.6 | 97.2 | 63.6 | 84.5 | 45.7 | 81.1 | 95.6 | 82.2 | 99.4 | 80.4 |
| Consensus TA (Wang et al., 2024) | 73.6 | 79.2 | 85.3 | 94.3 | 86.0 | 82.8 | 98.6 | 63.7 | 82.7 | 47.9 | 78.4 | 94.9 | 86.1 | 99.1 | 82.3 |
| TSV-M (Gargiulo et al., 2025) | 76.3 | 84.5 | 92.2 | 97.7 | 93.7 | 94.1 | 99.5 | 75.2 | 85.8 | 66.7 | 87.5 | 95.9 | 85.8 | 99.6 | 88.2 |
| ISO-CTS (Marczak et al., 2025a) | 79.2 | 88.6 | 95.1 | 98.7 | 91.9 | 96.9 | 99.4 | 80.6 | 88.2 | 62.1 | 96.5 | 96.1 | 82.5 | 99.5 | 89.7 |
| EMR-Merging (Huang et al., 2024) | 80.5 | 89.4 | 96.5 | 99.6 | 97.7 | 99.0 | 99.7 | 81.3 | 90.3 | 71.3 | 92.9 | 95.1 | 86.4 | 99.4 | 91.4 |
| AdaMerging (Yang et al., 2024b) | 76.0 | 91.1 | 92.1 | 97.0 | 93.5 | 97.4 | 98.9 | 77.4 | 82.5 | 50.8 | 89.9 | 95.7 | 51.2 | 99.2 | 85.2 |
| Surgery w/ Ada. (Yang et al., 2024a) | 77.9 | 91.2 | 94.7 | 98.4 | 94.4 | 98.2 | 99.3 | 81.5 | 84.6 | 71.4 | 95.4 | 95.9 | 83.7 | 99.5 | 90.4 |
| WEMoE (Tang et al., 2024b) | 80.7 | 92.6 | 95.3 | 98.8 | 96.7 | 98.6 | 99.7 | 83.2 | 91.4 | 44.7 | 95.8 | 96.2 | 62.2 | 99.3 | 88.4 |
| ProbSurgery w/ Ada. (Wei et al., 2025) | 77.5 | 90.9 | 95.1 | 98.7 | 94.6 | 98.6 | 99.3 | 81.4 | 85.0 | 71.6 | 96.7 | 96.1 | 85.3 | 99.5 | 90.7 |
| **SyMerge** | 81.2 | 92.0 | 96.8 | 99.7 | 97.0 | 99.0 | 99.4 | 84.8 | 91.3 | 76.2 | 98.5 | 95.7 | 88.7 | 99.5 | 92.8 |

Table N: Multi-task performance when merging ViT-L/14 models on 20 tasks.

| Method | SUN397 | Cars | RESISC45 | EuroSAT | SVHN | GTSRB | MNIST | DTD | CIFAR100 | FER2013 |
|---|---|---|---|---|---|---|---|---|---|---|
| Pretrained | 66.8 | 77.7 | 71.0 | 59.9 | 58.4 | 50.5 | 76.3 | 55.3 | 75.8 | 38.2 |
| Individual | 82.3 | 92.4 | 97.4 | 100.0 | 98.1 | 99.2 | 99.7 | 84.1 | 93.3 | 76.5 |
| Task arithmetic (Ilharco et al., 2023) | 71.2 | 76.2 | 78.0 | 79.3 | 75.3 | 65.7 | 95.9 | 59.7 | 82.5 | 42.0 |
| Ties Merging (Yadav et al., 2024) | 69.7 | 63.2 | 71.7 | 63.3 | 81.1 | 64.1 | 97.2 | 56.5 | 71.8 | 43.5 |
| PCB Merging (Du et al., 2024) | 68.2 | 56.5 | 70.2 | 68.4 | 81.3 | 68.1 | 97.6 | 56.5 | 68.1 | 43.8 |
| LiNeS w/ TA (Wang et al., 2025) | 72.1 | 77.6 | 80.6 | 84.6 | 76.5 | 69.0 | 96.5 | 62.0 | 83.3 | 43.5 |
| Consensus TA (Wang et al., 2024) | 73.7 | 75.9 | 83.2 | 84.8 | 82.2 | 71.8 | 97.7 | 61.8 | 82.1 | 45.5 |
| TSV-M (Gargiulo et al., 2025) | 74.9 | 79.7 | 89.9 | 95.7 | 90.3 | 90.4 | 97.9 | 72.8 | 83.3 | 60.1 |
| ISO-CTS (Marczak et al., 2025a) | 77.6 | 82.1 | 94.0 | 98.5 | 90.4 | 96.4 | 99.0 | 78.9 | 86.4 | 59.5 |
| EMR-Merging (Huang et al., 2024) | 79.5 | 88.9 | 95.9 | 99.3 | 96.9 | 98.7 | 99.6 | 79.3 | 90.1 | 64.6 |
| AdaMerging (Yang et al., 2024b) | 75.2 | 90.4 | 91.5 | 96.5 | 88.1 | 97.0 | 96.2 | 71.7 | 80.9 | 49.5 |
| Surgery w/ Ada. (Yang et al., 2024a) | 76.8 | 90.7 | 94.0 | 98.0 | 90.8 | 98.1 | 98.5 | 78.3 | 83.0 | 69.3 |
| WEMoE (Tang et al., 2024b) | 80.1 | 91.9 | 95.5 | 98.7 | 96.5 | 98.6 | 98.9 | 76.4 | 88.7 | 17.2 |
| ProbSurgery w/ Ada. (Wei et al., 2025) | 76.9 | 89.7 | 94.4 | 98.6 | 92.5 | 98.6 | 98.7 | 79.7 | 83.4 | 69.8 |
| **SyMerge** | 80.9 | 91.8 | 96.5 | 99.1 | 96.4 | 98.4 | 99.4 | 84.9 | 91.0 | 75.7 |

| Method | Flowers | Pet | PCAM | STL10 | CIFAR10 | EMNIST | FMNIST | Food101 | KMNIST | R-SST2 |
|---|---|---|---|---|---|---|---|---|---|---|
| Pretrained | 79.2 | 93.4 | 51.2 | 99.4 | 95.6 | 15.6 | 66.9 | 92.3 | 10.4 | 68.9 |
| Individual | 97.9 | 95.2 | 90.0 | 99.3 | 99.3 | 99.8 | 96.0 | 95.4 | 98.7 | 85.6 |
| Task arithmetic | 71.2 | 76.2 | 78.0 | 79.3 | 75.3 | 65.7 | 95.9 | 59.7 | 82.5 | 42.0 |
| Ties Merging | 67.5 | 93.7 | 70.3 | 97.4 | 94.8 | 85.7 | 83.5 | 82.0 | 34.9 | 67.1 |
| PCB Merging | 74.0 | 93.2 | 75.9 | 97.0 | 93.8 | 85.9 | 83.9 | 78.2 | 37.4 | 69.5 |
| LiNeS w/ TA | 72.1 | 77.6 | 80.6 | 84.6 | 76.5 | 69.0 | 96.5 | 62.0 | 83.3 | 43.5 |
| Consensus TA | 76.4 | 95.1 | 79.4 | 99.0 | 97.5 | 85.2 | 91.3 | 38.6 | 71.2 |  |
| TSV-M | 85.3 | 96.0 | 86.0 | 99.4 | 98.1 | 98.8 | 90.9 | 92.4 | 79.7 | 82.8 |
| ISO-CTS | 94.1 | 95.4 | 82.8 | 99.4 | 98.3 | 95.6 | 92.3 | 93.2 | 89.2 | 82.3 |
| EMR-Merging | 96.5 | 95.7 | 87.1 | 99.5 | 99.0 | 99.8 | 94.5 | 94.1 | 95.1 | 85.1 |
| AdaMerging | 88.4 | 95.5 | 50.4 | 99.0 | 97.0 | 97.0 | 91.5 | 92.1 | 10.0 | 83.9 |
| Surgery w/ Ada. | 94.2 | 95.7 | 83.3 | 99.4 | 97.9 | 99.1 | 92.3 | 92.4 | 72.7 | 85.6 |
| WEMoE | 98.3 | 96.0 | 51.2 | 99.4 | 98.0 | 99.1 | 37.4 | 94.8 | 10.0 | 49.9 |
| ProbSurgery w/ Ada. | 95.8 | 95.1 | 83.9 | 99.4 | 98.1 | 99.2 | 92.7 | 92.8 | 80.3 | 85.0 |
| **SyMerge** | 98.4 | 95.7 | 88.0 | 99.4 | 98.6 | 99.2 | 93.7 | 93.8 | 97.9 | 85.5 |

