# OpenReview forum: "SyMerge: From Non-Interference to Synergistic Merging via Single-Layer Adaptation"
_ICLR.cc/2026/Conference — Submitted to ICLR 2026_

### Official Review · Reviewer_KP85 · 2025-10-17

**Soundness:** 3
**Presentation:** 3
**Contribution:** 2
**Rating:** 4
**Confidence:** 3

**Summary:**

The authors' motivation stems from an empirical finding that a model's cross-task generalization capability is a strong predictor of its merging performance.

SyMerge is a lightweight, test-time adaptation method that works with unlabeled data. It jointly optimizes two components: the merging coefficients for the shared encoder and a single task-specific layer (e.g., the classifier). To provide a stable training signal without ground-truth labels, it employs a robust self-labeling strategy, using the predictions from the individual fine-tuned "expert" models as targets.

**Strengths:**

Clarity: The motivation is clearly laid out in Section 2.2, with intuitive figures that build a strong case for the authors' approach. The methodology is described precisely, and the connection between the motivating pilot study and the final SyMerge design is logical and clear.

Significance: Model merging is an efficient and increasingly important alternative to full multi-task training. By providing a method that is not only effective but also lightweight and scalable, the paper offers a practical solution to a relevant problem. SyMerge consistently outperforms other methods, often by a large margin, especially as the number of tasks increases (Table 1). Furthermore, the insight that the adapted layers are transferable (Table 4) opens up interesting possibilities for improving existing model merging techniques.

**Weaknesses:**

SyMerge's approach involves jointly training the merging coefficients and a task-specific layer. However, for the comparison to be entirely fair, it is crucial to understand if the baseline methods are afforded a similar adaptation step. A significant portion of its performance improvement could be attributed to this classifier fine-tuning, rather than purely to the superiority of the merged encoder's representations.

The theoretical justification in Section 3.2 hinges on the assumption of "cross-task linearity". While this is a reasonable starting point for analysis, it is a strong assumption that may not fully capture the complex, non-linear interactions within deep neural networks. The paper would be strengthened if the authors could include a brief discussion on the limitations of this assumption or provide some empirical validation suggesting it holds approximately in their experimental settings.

Confidence-based filtering mechanism seems to be used for the vision tasks. A small ablation in the main paper showing the performance impact of this filtering mechanism would be helpful to clarify whether it is a minor tweak or a critical component for achieving the reported results.

**Questions:**

In lines 144-149, you define two performance metrics: (1) "cross-task performance" (Encoder A + Classifier B) and (2) "merging performance" (merged A&B encoder + Classifier B). Could you clarify why the latter is considered "merging performance" when it's evaluated only on Task B? Is it simply an average over all possible B's for a given set of models?

---

> ### Author Response · Authors · 2025-11-25
> **Response to the Reviewer KP85 [1/3]**
>
> Dear Reviewer `KP85`, The authors sincerely appreciate your thoughtful and detailed evaluation of our work. We are grateful that you highlighted the clarity of the motivation and methodology, as well as the practical significance and strong empirical performance of SyMerge. Your comments on the role of adaptation, the strength of the cross task linearity assumption, and the effect of the confidence based filtering mechanism have been particularly helpful in guiding our revisions. We also thank you for your clear question regarding the definition of merging performance. We address all of your comments and questions in the responses below.
>
>
>
>
> | **W1:** However, for the comparison to be entirely fair, it is crucial to understand if the baseline methods are afforded a similar adaptation step.  |
> | :-
>
>
> To address this concern, we conducted an additional ablation in which we also allowed several representative baselines to adapt their classifier heads using the same unlabeled data and training budget. This adaptation indeed improves their performance, but **SyMerge still achieves higher accuracy** under the same conditions, indicating that **our gains do not stem solely from having an extra classifier adaptation stage**.
>
> | Encoder        | Avg. (with SyMerge) |
> |----------------|----------------------|
> | Pretrained     | 81.9                 |
> | Task Arithmetic| 88.2                 |
> | AdaMerging     | 89.6                 |
> | Ours           | 90.1                 |
>
> We also **respectfully believe that our comparison is not unfair**. Each **baseline uses its its own test-time adaptation or merging procedure**, many of which either cannot be straightforwardly integrated into our framework or incur high computational cost. In contrast, SyMerge achieves competitive performance with a relatively lightweight adaptation step.
>
>
> | **W2:** A significant portion of its performance improvement could be attributed to this classifier fine-tuning, rather than purely to the superiority of the merged encoder's representations. |
> | :-
>
> We agree that classifier fine-tuning can contribute to the overall performance. **To separate this effect from improvements** in the merged encoder, we conducted an additional analysis.
> (1) We first trained SyMerge in the standard way.
> (2) After training, we **removed the learned classifier**, restored the **original zero-shot classifier**, and evaluated the model again.
>
> This model still **reached an average accuracy of 82.4%,** which is a **large improvement over the Task Arithmetic** initialization using the same zero-shot classifier **(69.1%)**. Because the classifier is identical in both evaluations, this result shows that **SyMerge also enhances the underlying merged encoder**, and that the performance gains cannot be attributed solely to classifier fine-tuning.
>
> We have added these results in the revised manuscript (Section 4.3 - Improvement beyond classifier adaptation).

---

> ### Author Response · Authors · 2025-11-25
> **Response to the Reviewer KP85 [2/3]**
>
> | **W3:** Empirical evidence for CTL: The paper would be strengthened if the authors could include a brief discussion on the limitations of this assumption or provide some empirical validation suggesting it holds approximately in their experimental settings. |
> | :-
>
>
> Following your suggestion, we have provided an empirical sanity check that it holds approximately in our setting.
>
> Concretely, we consider the CTL relation $f(x; \theta(\alpha)) \approx (1-\alpha) f(x; \theta_i) + \alpha f(x; \theta_j)$, where $f$ is the vision encoder, $\theta(\alpha) = (1-\alpha)\theta_i + \alpha\theta_j$ denotes a linear interpolation of parameters between two task-specific models $i$ and $j$, and $\alpha \in [0, 1]$. If CTL holds, the representation $f(x; \theta(\alpha))$ should stay closely aligned with the corresponding interpolation of $f(x; \theta_i)$ and $f(x; \theta_j)$ across $\alpha$.
>
> To test this, we take SUN397 as the anchor task and examine CTL with respect to the other seven tasks. For each task pair $(\text{SUN397}, j)$, and for multiple values of $\alpha$ between 0 and 1, we compute the cosine similarity between $f(x; \theta(\alpha))$ and the interpolated representation $(1-\alpha) f(x; \theta_{\text{SUN}}) + \alpha f(x; \theta_j)$.
>
> As shown in the table, **the cosine similarity is consistently high and close to 1 in most cases**, indicating that **CTL is a reasonable approximate** description of the encoder behavior in our experimental regime.
>
> In the revised version, we have incorporated this quantitative analysis along with its discussion in Appendix D.4, clearly demonstrating that it provides empirical support for the CTL assumption.
>
>
> | Coefficient | 0.1 | 0.2 | 0.3 | 0.4 | 0.5 | 0.6 | 0.7 | 0.8 | 0.9 |
> |-------------|-----|-----|-----|-----|-----|-----|-----|-----|-----|
> | Cars        | 1.00 | 0.99 | 0.98 | 0.97 | 0.97 | 0.97 | 0.98 | 0.99 | 1.00 |
> | RESISC45    | 0.99 | 0.98 | 0.97 | 0.96 | 0.95 | 0.95 | 0.96 | 0.98 | 0.99 |
> | EuroSAT     | 0.99 | 0.98 | 0.96 | 0.95 | 0.94 | 0.94 | 0.95 | 0.97 | 0.99 |
> | SVHN        | 0.99 | 0.97 | 0.94 | 0.90 | 0.87 | 0.85 | 0.87 | 0.92 | 0.97 |
> | GTSRB       | 0.99 | 0.96 | 0.93 | 0.90 | 0.88 | 0.88 | 0.90 | 0.94 | 0.98 |
> | MNIST       | 0.99 | 0.97 | 0.94 | 0.92 | 0.90 | 0.89 | 0.91 | 0.95 | 0.98 |
> | DTD         | 0.99 | 0.98 | 0.96 | 0.96 | 0.96 | 0.96 | 0.97 | 0.99 | 1.00 |

---

> ### Author Response · Authors · 2025-11-25
> **Response to the Reviewer KP85 [3/3]**
>
> | **W4:** A small ablation in the main paper showing the performance impact of this filtering mechanism would be helpful to clarify whether it is a minor tweak or a critical component for achieving the reported results. |
> | :-
>
> In response to your suggestion, we have run an additional ablation in which we disable the confidence-based filtering while keeping all other settings identical. As shown in the table below, **confidence filtering brings a consistent but modest performance improvement** (reported as mean ± std over 5 runs), while SyMerge **without filtering still clearly outperforms existing merging baselines**.
>
> In the revised manuscript, we have included these results in Appendix D.5 and in Table F.
>
> | Tasks    | Model     | Mean (No Filtering) | Std (No Filtering) | Mean (With Filtering) | Std (With Filtering) |
> |----------|-----------|----------------------|---------------------|-------------------------|------------------------|
> | 8 tasks  | ViT Base  | 89.7                | 0.1                 | 90.1 (+0.4)             | 0.1                    |
> | 14 tasks | ViT Base  | 88.3                | 0.2                 | 88.7 (+0.4)             | 0.1                    |
> | 20 tasks | ViT Base  | 88.5                | 0.3                 | 88.6 (+0.1)             | 0.4                    |
> | 8 tasks  | ViT Large | 93.9                | 0.0                 | 94.1 (+0.2)             | 0.0                    |
>
>
>
> | **Q1:** In lines 144-149, you define two performance metrics: (1) "cross-task performance" (Encoder A + Classifier B) and (2) "merging performance" (merged A&B encoder + Classifier B). Could you clarify why the latter is considered "merging performance" when it's evaluated only on Task B? |
> | :-
>
> Thank you for pointing out this ambiguity. In our pilot study, we treat each task B as a reference task and define the two metrics with respect to B.
> For a fixed task B, we consider all other tasks A (A ≠ B). We then compute
> (1) **the cross-task performance of B** as the average accuracy of "Encoder A + Classifier B" on task B, averaged over all such A, and
> (2) **the merging performance of B** as the average accuracy of "merged(Encoder A, Encoder B) + Classifier B" on task B, again averaged over all such A.
>
> We refer to (2) as "**merging performance**" because it measures how well task B performs after its encoder has been merged with encoders from other tasks. In other words, this metric directly reflects how much task interference from other tasks degrades or preserves the performance on B, and thus serves as an indicator of the quality of the merging procedure for that task.
>
> We have clarified this definition in the revised version (Section 2.2 Motivation - Rethinking cross-task performance).

---

### Official Review · Reviewer_m1Pr · 2025-10-29

**Soundness:** 3
**Presentation:** 3
**Contribution:** 3
**Rating:** 4
**Confidence:** 4

**Summary:**

The authors propose SyMerge, a lightweight model merging framework that jointly optimizes task-specific layers and merging coefficients at test time, shifting the goal from "avoiding interference" to actively seeking synergy between tasks. The core idea stems from a set of pilot experiments: cross-task performance of encoders from different tasks strongly predicts merging quality; and fine-tuning only a single task layer significantly improves this compatibility. To ensure stable unsupervised adaptation, the authors abandon unstable entropy minimization and instead use self-labeling guided by predictions from various experts. The paper demonstrates strong quantitative and ablation results on multiple benchmarks in vision, dense prediction, and NLP.

**Strengths:**

1. The paper offers a novel objective and perspective, shifting from non-interference to synergy, providing a clear conceptual framework and explaining why some merging methods have upper bounds.
2. The method is concise and efficient—it jointly optimizes a single layer and coefficients only during testing, without introducing additional large models or modules, resulting in low engineering implementation costs.
3. The writing is quite easy to read and it was well-written

**Weaknesses:**

1. The method essentially uses the predictions of individual models as supervision signals. When some experts are systematically inaccurate in the target domain, spurious labels may guide the merged model towards incorrect solutions. This point is mentioned in the limitation section of the paper, but it lacks quantitative sensitivity analysis for low-quality experts.
2. Although the paper provides default learning rate, iteration count, and initialization, it lacks hyperparameter sensitivity curves and computational costs for different task numbers or model scales. This will affect the usability evaluation of the method in real-world large-scale scenarios.
3. The theoretical conditions are relatively strong; Proposition 1 is based on the assumptions of "cross-task linearity" and convex output loss. However, the nonlinearity of actual depth models may render these assumptions incomplete. The paper provides a proof, but lacks empirical testing of the approximation of these assumptions on the used benchmark.
4. The method requires a small amount of unlabeled target domain data. However, target domain data is difficult to obtain for many tasks.

**Questions:**

1. The method requires a small amount of unlabeled target domain data. However, target domain data is difficult to obtain for many tasks. How sensitive is the method to this target domain data? Can it be extended to scenarios with no data or very little data?
2. How computational resources and efficiency does the method offer for every stage?
3. Is the improvement in the method due to the synergy brought about by merging or the result of additional data alignment? If we don't use task vectors and only use the base model, can we also improve the model performance through this additional target domain data? Introducing target domain data is unfair to comparing with other methods; can other methods also achieve better results by introducing this target domain data? The authors need to conduct further research.
4. How does the method perform on larger models? How can target domain data be used for collaboration on some NLP generation tasks?

If the author can solve the question and the weakness well, i will raise my score.

---

> ### Author Response · Authors · 2025-11-25
> **Response to the Reviewer m1Pr [1/5]**
>
> Dear Reviewer m1Pr, The authors sincerely appreciate your thoughtful and detailed review. Your recognition of the strengths of our work, including the novel objective, the synergy-driven perspective, and the efficiency of the proposed method is greatly encouraging. At the same time, your constructive comments on the limitations, such as sensitivity to low-quality experts, hyperparameter analysis, theoretical assumptions, and reliance on target-domain data, as well as the insightful questions you raised across methodological and empirical aspects, have been invaluable in helping us further refine and strengthen the paper. We address all of your concerns in detail in the following responses.
>
>
> | **W1:** When some experts are systematically inaccurate in the target domain, spurious labels may guide the merged model towards incorrect solutions. This point is mentioned in the limitation section of the paper, but it lacks quantitative sensitivity analysis for low-quality experts. |
> | :-
>
> Our method already incorporates a mechanism to mitigate noisy predictions: low-confidence samples are removed during self-labeling, which **reduces the influence of unreliable experts**. As shown in the table below, enabling this filtering consistently improves performance (mean ± std over 5 runs). We agree that understanding how to handle systematically inaccurate experts, and how to strengthen such experts through adaptation, is an important direction for future work.
>
> We have added these results in Appendix D.5 and Table F of our revised manuscript.
>
>
> | Tasks    | Model     | Mean (No Filtering) | Std (No Filtering) | Mean (With Filtering) | Std (With Filtering) |
> |----------|-----------|----------------------|---------------------|-------------------------|------------------------|
> | 8 tasks  | ViT Base  | 89.7                | 0.1                 | 90.1 (+0.4)             | 0.1                    |
> | 14 tasks | ViT Base  | 88.3                | 0.2                 | 88.7 (+0.4)             | 0.1                    |
> | 20 tasks | ViT Base  | 88.5                | 0.3                 | 88.6 (+0.1)             | 0.4                    |
> | 8 tasks  | ViT Large | 93.9                | 0.0                 | 94.1 (+0.2)             | 0.0                    |

---

> ### Author Response · Authors · 2025-11-25
> **Response to the Reviewer m1Pr [2/5]**
>
> | **W2:** Although the paper provides default learning rate, iteration count, and initialization, it lacks hyperparameter sensitivity curves and computational costs for different task numbers or model scales. |
> | :-
>
> To verify real-world usability, we expanded our analysis to ViT-L/14 and 20 tasks.
>
> **1. Sensitivity & Convergence**
> As shown in the tables below, SyMerge demonstrates rapid convergence (reaching near-optimal performance within 100 iterations) and high stability across a wide range of learning rates, even in large-scale settings.
>
> **2. Initialization**
> Our additional experiments on ViT-L/14 (presented below) confirm robust performance regardless of initialization strategies.
>
> We have added these results in Appendix D.3 and Figure E,F of our revised manuscript.
>
> `ViT Large – 8 tasks`
> | Learning Rate     | 100  | 200  | 300  | 400  | 500  |
> |-------------------|------|------|------|------|------|
> | 5.00E-02          | 91.6 | 92.1 | 92.7 | 92.6 | 93.0 |
> | 1e-2 (baseline)   | 93.1 | 93.6 | 93.8 | 94.0 | 94.1 |
> | 5.00E-03          | 93.0 | 93.7 | 93.8 | 94.0 | 94.1 |
> | 1.00E-03          | 90.9 | 92.1 | 92.7 | 93.1 | 93.3 |
>
> `ViT Base – 20 tasks`
> | Learning Rate     | 100  | 200  | 300  | 400  | 500  |
> |-------------------|------|------|------|------|------|
> | 5.00E-02          | 79.9 | 80.4 | 83.5 | 83.1 | 83.0 |
> | 1e-2 (baseline)   | 86.8 | 87.6 | 88.0 | 88.7 | 88.6 |
> | 5.00E-03          | 86.4 | 87.5 | 87.9 | 88.9 | 89.0 |
> | 1.00E-03          | 82.4 | 85.3 | 86.3 | 87.2 | 87.6 |
>
> `ViT Large – 20 tasks`
> | Learning Rate     | 100  | 200  | 300  | 400  | 500  |
> |-------------------|------|------|------|------|------|
> | 5.00E-02          | 88.6 | 89.6 | 90.1 | 90.3 | 90.6 |
> | 1e-2 (baseline)   | 92.1 | 92.7 | 93.1 | 93.3 | 93.6 |
> | 5.00E-03          | 92.1 | 92.8 | 93.2 | 93.4 | 93.6 |
> | 1.00E-03          | 89.3 | 91.1 | 91.9 | 92.4 | 92.6 |
>
> `ViT-Large - 8 tasks on different initializations`
> |Initialization| Trainable | 100  | 200  | 300  | 400  | 500  |
> |--------------|-----------|------|------|------|------|------|
> |Task arithmetic|CLS + Coef| 93.1 | 93.6 | 93.8 | 94.0 | 94.1 |
> |Task arithmetic|  CLS     | 92.0 | 92.0 | 92.1 | 92.1 | 92.0 |
> |AdaMerging     |CLS + Coef| 93.3 | 93.6 | 93.9 | 94.0 | 94.0 |
> |AdaMerging     |  CLS     | 92.0 | 92.0 | 92.1 | 92.1 | 92.1 |
>
>
>
>
>
> | **W2 & Q2:** How computational resources and efficiency does the method offer for every stage?|
> | :-
>
> To evaluate efficiency, we measured peak GPU memory and trainable parameters on 8 vision tasks. We standardized all methods to employ sequential updates (task-by-task) with a batch size of 16, incorporating lightweight implementation-level optimizations for memory management. This adaptation was necessary because the original simultaneous update schemes of AdaMerging and WeMoE incur excessive memory costs, leading to Out-Of-Memory (OOM) errors on ViT-L/14.
>
> As shown in the table below, SyMerge demonstrates high efficiency. Its memory usage is comparable to the most lightweight baseline (AdaMerging), while introducing negligible trainable parameters. This confirms that SyMerge achieves state-of-the-art performance with a minimal computational footprint.
>
> We have added these results in Appendix D.6 and Table G of our revised manuscript.
>
>
> | Method      | GPU Mem (B/32) | GPU Mem (L/14) | Params (B/32) | Params (L/14) |
> |-------------|----------------|----------------|----------------|----------------|
> | Ours        | 7.12GB         | 23.71GB        | 0.39M          | 0.59M          |
> | AdaMerging  | 7.09GB         | 22.88GB        | 0.00M          | 0.00M          |
> | Surgery     | 7.13GB         | 30.44GB        | 0.13M          | 0.20M          |
> | WEMoE       | 8.01GB         | 33.57GB        | 7.16M          | 25.39M         |

---

> ### Author Response · Authors · 2025-11-25
> **Response to the Reviewer m1Pr [3/5]**
>
> | **W3:** The theoretical conditions are relatively strong; Proposition 1 is based on the assumptions of "cross-task linearity" and convex output loss. However, the nonlinearity of actual depth models may render these assumptions incomplete. The paper provides a proof, but lacks empirical testing of the approximation of these assumptions on the used benchmark. |
> | :-
>
> We clarify our assumptions and provide supporting evidence below.
>
> ### **1. Empirical Verification of Cross-Task Linearity (CTL)**
> To validate the Cross-Task Linearity (CTL) assumption bridging parameter-space merging and representation ensembling, we conducted the suggested quantitative analysis. We compared the output of the parameter-interpolated model, $f(x; (1-\alpha)\theta_i + \alpha\theta_j)$, against the functional interpolation, $(1-\alpha)f(x; \theta_i) + \alpha f(x; \theta_j)$.
>
> Measuring cosine similarity across task pairs (anchor: SUN397) with varying $\alpha \in [0, 1]$, **we observed consistently high similarity scores** (see Table below). This confirms that models fine-tuned from a common pre-trained backbone exhibit strong linearity within the shared solution basin, **validating CTL as a robust approximation in our experimental regime**.
>
> We have included this quantitative analysis and its discussion in Appendix D.4 of our revised manuscript, explicitly clarifying that it provides empirical support for the CTL assumption.
>
> | Coefficient | 0.1 | 0.2 | 0.3 | 0.4 | 0.5 | 0.6 | 0.7 | 0.8 | 0.9 |
> |-------------|-----|-----|-----|-----|-----|-----|-----|-----|-----|
> | Cars        | 1.00 | 0.99 | 0.98 | 0.97 | 0.97 | 0.97 | 0.98 | 0.99 | 1.00 |
> | RESISC45    | 0.99 | 0.98 | 0.97 | 0.96 | 0.95 | 0.95 | 0.96 | 0.98 | 0.99 |
> | EuroSAT     | 0.99 | 0.98 | 0.96 | 0.95 | 0.94 | 0.94 | 0.95 | 0.97 | 0.99 |
> | SVHN        | 0.99 | 0.97 | 0.94 | 0.90 | 0.87 | 0.85 | 0.87 | 0.92 | 0.97 |
> | GTSRB       | 0.99 | 0.96 | 0.93 | 0.90 | 0.88 | 0.88 | 0.90 | 0.94 | 0.98 |
> | MNIST       | 0.99 | 0.97 | 0.94 | 0.92 | 0.90 | 0.89 | 0.91 | 0.95 | 0.98 |
> | DTD         | 0.99 | 0.98 | 0.96 | 0.96 | 0.96 | 0.96 | 0.97 | 0.99 | 1.00 |
>
>
>
>
>
> ### **2. Robustness to Non-Convexity (Disjoint Basins)**
> The reviewer rightfully points out that the linearity assumption may not hold in complex deep learning landscapes. To rigorously test our method's efficacy when this **linearity assumption breaks down completely** (i.e., extreme non-linearity or disjoint basins), we **conducted an additional stress test merging models with different initializations**.
>
> We merged a DTD expert ($W_1$) fine-tuned from OpenAI-ViT and a EuroSAT expert ($W_2$) from DataComp-ViT. Unlike standard setups, these models lie in disjoint basins, creating a severe non-linear loss barrier between them. Since there is no shared pre-trained weight, we defined task vectors relative to the weight average ($W_{avg}$).
>
> As shown in the table below, standard **Weight Averaging collapses**, confirming that the linearity assumption does not hold in this setting due to loss barrier. **Surgery also fails** as it relies on the frozen, misaligned backbone. In contrast, SyMerge successfully recovers performance.
> This experiment empirically demonstrates that **SyMerge remains effective even when the theoretical assumption of linearity is severely compromised**, validating its practical robustness beyond the idealized theoretical bounds.
>
> We have included this stress-test experiment and its results in the revised manuscript (Section 4.3 Empirical Analyses - Merging across disjoint basins, Table 5).
>
> |                     | EuroSAT | DTD  | AVG  |
> |---------------------|---------|------|------|
> | Individual (Euro)   | 98.1    | 3.6  | 50.8 |
> | Individual (DTD)    | 2.0     | 79.4 | 40.7 |
> | Weight Average      | 11.6    | 2.2  | 6.9  |
> | AdaMerging          | 8.6     | 2.1  | 5.4  |
> | Surgery             | 38.1    | 13.1 | 25.6 |
> | Ours                | 96.2    | 62.0 | 79.1 |

---

> ### Author Response · Authors · 2025-11-25
> **Response to the Reviewer m1Pr [4/5]**
>
> | **W4 & Q1:** Data availability|
> | :-
>
> To address the concerns regarding data availability, we conducted a comprehensive analysis covering scenarios from limited data to single-sample streaming.
>
>
> ### **1. Online Test-Time Adaptation**
> To demonstrate practicality in challenging regimes where data is not available in batches, we evaluated an Online Test-Time Adaptation setting. At each step, **the model receives a single unlabeled test sample**, updates its parameters, and is evaluated on that sample immediately.
>
> As shown in Table below, **SyMerge adapts quickly to the stream of single-sample inputs**. It consistently **outperforms other TTA** approaches and clearly surpasses training-free baselines such as Task Arithmetic (69.1), Ties Merging (72.9), LiNeS (74.1), and Consensus TA (74.9). This result proves that SyMerge is viable even in streaming scenarios where data accumulation is difficult.
>
>
>
> | SEEN SAMPLES | 10   | 20   | 50   | 100  | 200  | 500  |
> |--------------|------|------|------|------|------|------|
> | AdaMerging   | 70.0 | 71.5 | 71.1 | 72   | 72.6 | 74.1 |
> | Surgery      | 70.0 | 71.1 | 70.9 | 73.4 | 73.2 | 73.6 |
> | Ours         | 73.2 | 75.3 | 75.7 | 77.1 | 80   | 82.1 |
>
>
>
> ### **2. Sensitivity to Data Quantity**
> We also evaluated SyMerge's performance when training from weights initialized by Task Arithmetic, using **varying amounts of unlabeled test data** (1%, 5%, 10%, and 20% of the test set, with 250 iterations). As shown in the table below, SyMerge outperforms many training-free merging methods, such as Task Arithmetic (69.1), Ties Merging (72.9), LiNeS (74.1), and Consensus TA (74.9), as well as test-time adaptive methods, even with extremely limited data.
>
>
> | Model     | 1%   | 5%   | 10%  | 20%  | 100% |
> |-----------|------|------|------|------|------|
> | AdaMerging       | 76.0 | 77.2 | 77.5 | 77.7 | 80.1 |
> | Surgery   | 72.8 | 77.3 | 78.4 | 78.9 | 80.9 |
> | Ours | 78.1 | 84.5 | 86.7 | 88.0 | 90.1 |
>
>
> In the revised version, We have clarified that these experiments directly address robustness to the size of the unlabeled test data (Appendix B.5. Robustness to Data Scarcity).

---

> ### Author Response · Authors · 2025-11-25
> **Response to the Reviewer m1Pr [5/5]**
>
> | **Q3-1:** Is the improvement in the method due to the synergy brought about by merging or the result of additional data alignment?  |
> | :-
>
> We clarify that our method operates **under the standard Test-Time Adaptation (TTA) protocol**, using the same unlabeled test stream as other TTA baselines, **without any additional proxy data**. Therefore, the answer to this question is "no, the improvement is **not simply the result of additional data alignment**.".
>
> Furethermore, to prove that our improvement **stems from unlocking synergy** rather than mere data alignment, we highlight the results from our Online (Continual) Test-Time Adaptation experiment. As shown in the Online TTA table before (refer to W4 & Q1: Data availability), SyMerge quickly surpasses training-free methods (e.g., Task Arithmetic , Ties-Merging ) even with single-sample updates. This indicates that **minimal alignment is sufficient to trigger positive synergy** that static merging fails to capture.
>
>
>
>
> | **Q3-2:** If we don't use task vectors and only use the base model, can we also improve the model performance through this additional target domain data?   |
> | :-
>
> We applied our method to the pre-trained base model using target data without task vectors. While accuracy reached 81.9%, it notably falls short of SyMerge (90.1%). This gap confirms that target data alone is insufficient without task vectors. Thus, **SyMerge's superiority stems from the synergy between task-specific knowledge and our adaptation**, not merely from data access.
>
>
>
> | **Q3-3:** Can other methods also achieve better results by introducing this target domain data?  |
> | :-
>
>
> We would like to clarify that **once target data is accessible**, the **method no longer belongs to the 'training-free'** catergory but instead falls **under the 'test-time adaptation'** category. For this reason, our evaluation already compares SyMerge against state-of-the-art test-time adaptation approaches that also leverage unlabeled target data, including AdaMerging, Surgery and WeMoE. As shown in Table 1, SyMerge significantly outperforms these baselines, demonstrating that **our superiority stems from our specific adaptation methodology** (joint optimization and expert-guided self-labeling), not merely from access to target data.
>
> To further support this point, we examined whether target-domain data alone can improve existing merging methods. We applied SyMerge's adaptation step on top of the fixed encoder produced by Task Arithmetic. Despite the same target data being available, Task Arithmetic achieved only 69.1%, whereas applying SyMerge lifted performance to 88.2%. This confirms that while target data offers potential, SyMerge is key to effectively unlocking it.
>
>
>
>
>
>
>
> | **Q4:** Larger model, NLP Generation tasks |
> | :-
>
>
> To examine scalability, we evaluated SyMerge on a larger sequence-to-sequence model, FLAN-T5 Base, across eight GLUE tasks cast in a text-generation format. We compared four settings: training individual experts, Task Arithmetic, AdaMerging, and SyMerge. The average scores are as follows:
> - Individual: 86.4
> - Task Arithmetic: 78.9
> - AdaMerging: 81.5
> - Ours: 83.5
>
> SyMerge consistently improves over Task Arithmetic and AdaMerging in this larger LM setting, while narrowing the gap to individually trained experts. Generation models also output token-level probability distributions, so our method applies directly by matching the merged model’s distribution to each expert’s distribution on unlabeled target text using cross-entropy.

---

### Official Review · Reviewer_3JoF · 2025-10-31

**Soundness:** 3
**Presentation:** 3
**Contribution:** 2
**Rating:** 6
**Confidence:** 4

**Summary:**

The paper introduces SyMerge, a model merging framework that pursues not just the avoidance of task interference but active task synergy when merging independently fine-tuned models. The method achieves this by jointly adapting a single task-specific layer together with merging coefficients at test time, guided by expert model self-labels, rather than unstable entropy minimization. The authors provide both theoretical and empirical evidence for why enhancing cross-task compatibility is key for successful model merging and demonstrate SyMerge’s effectiveness across vision, dense prediction, and NLP tasks.

**Strengths:**

1. The work advances a shift in objectives for model merging—arguing for positive synergy rather than mere non-interference. This reconceptualization is original in the landscape of model merging.
2. Theoretical justification is provided showing that improved cross-task performance tightens loss bounds for merged models, supporting the focus on functional alignment.
3.  SyMerge outperforms a strong suite of prior model merging baselines in multi-task classification, dense prediction, and NLP, with results approaching those of individually fine-tuned models.

**Weaknesses:**

1. While the pursuit of task synergy is motivated well, the core adaptation step (jointly tuning a single layer and coefficients with self-labeling) is a fairly incremental extension over test-time adaptive methods such as AdaMerging. The framework design—minimizing cross-entropy or L1 to match expert predictions—can be considered a straightforward application of self-labeling in existing frameworks (Representation Surgery and WUDI-Merging).
2. The proposed method’s reliance on the predictions from the individual expert models as supervision means that improvements are ultimately bounded by the expert’s limitations.
3. The theoretical analysis relies on known assumptions such as cross-task linearity and convexity, but practical models do not strictly satisfy these. The proof glosses over how close “approximate” linearity is achieved in practice, and no bounds or ablations connect the assumption to observed efficacy.

**Questions:**

1. Can the authors provide empirical analysis on the impact of the cross-task linearity assumption underpinning Proposition 1? Are there cases where nonlinear interactions or loss non-convexity cause SyMerge to underperform?

---

> ### Author Response · Authors · 2025-11-25
> **Response to the Reviewer 3JoF [1/3]**
>
> Dear Reviewer `3JoF`,
> The authors sincerely appreciate your thoughtful evaluation of our work. We are grateful that you recognized the novelty of our objective shift toward positive task synergy, the theoretical motivation supporting functional alignment, and the strong empirical performance across diverse tasks. Your constructive comments regarding the methodological incrementalism, reliance on expert predictions, and the assumptions underlying our theoretical analysis were highly valuable. These insights helped us more clearly understand the limitations of our framework and guided us to conduct additional analyses. We address your questions and concerns in detail in the responses below.
>
>
> | **W1:** While the pursuit of task synergy is motivated well, the core adaptation step (jointly tuning a single layer and coefficients with self-labeling) is a fairly incremental extension over test-time adaptive methods  |
> | :-
>
> We **respectfully disagree that SyMerge is a straightforward or incremental extension** of prior test-time adaptive methods. Although we share high-level elements such as self-labeling, **our joint optimization framework resolves fundamental failure** modes that methods like AdaMerging, Representation Surgery, and WUDI-Merging cannot address.
>
> ### **1. Architectural differences**
> AdaMerging optimizes only the merging coefficients, which often fails to resolve deeper feature-level conflicts (Table 5 of revised manuscript: Coeff-only 84.6% vs. SyMerge 90.1%). Surgery adds external adapters on top of a frozen merged encoder, so it cannot repair misaligned internal representations.
> SyMerge jointly tunes **both the coefficients and an internal layer**, allowing the shared encoder and task-specific heads to adjust to each other. This **creates cross-task alignment** that neither coefficients-only nor adapter-based approaches can achieve.
>
>
>
> ### **2. Handling incompatibilities**
> SyMerge is also effective in situations where existing merging or distillation-based methods fail due to task-level or parameter-level mismatch.
>
> - **Task-level mismatch**: In Table 3, CoLA focuses on syntactic acceptability and differs substantially from the semantic tasks in GLUE. Prior merging methods degrade sharply on this task, and even distillation-based approaches such as Representation Surgery (46.8) and ProbSurgery [1] (54.7) cannot recover the expert's performance (60.2). **SyMerge (60.0) remains stable despite this heterogeneity**, indicating that the joint adaptation resolves functional conflicts that parameter-level merging cannot address.
>
> | Method   | Individual | Task Arithmetic   | Representation Surgery | ProbSurgery | SyMerge   |
> |---------|-----------:|----------------:|--------:|-----------:|----------:|
> | `CoLA`    | 60.2       | 18.8             | 46.8    | 54.7        | **60.0**  |
>
> - **Parameter-level mismatch**: We also evaluated experts fine-tuned from different pretrained checkpoints (OpenAI-ViT vs. DataComp-ViT), a setting where the backbone parameters are fundamentally misaligned. Baselines such as **Surgery collapse** under this mismatch, while **SyMerge restores performance** by adjusting both the coefficients and the adapted layer. This shows that SyMerge can realign internal representations even when the starting parameters differ significantly. (**See our response to Q2 for details**.)
>
> `Reference`
> [1] *Representation Surgery in Model Merging with Probabilistic Modeling*, ICML 2025
>
>
> | **W2:** The proposed method’s reliance on the predictions from the individual expert models as supervision means that improvements are ultimately bounded by the expert’s limitations. |
> | :-
>
> In model merging, individual experts are typically regarded as the **empirical upper bound**, and recovering their full performance is difficult due to task interference. SyMerge narrows this gap substantially and achieves performance that existing methods often fail to reach. It also promotes positive cross-task synergy, as shown in Table 4, improving functional alignment rather than degrading any expert's capabilities. Although surpassing the experts themselves is beyond current merging frameworks, **SyMerge is a meaningful step toward preserving and integrating their strengths** within a unified model.

---

> ### Author Response · Authors · 2025-11-25
> **Response to the Reviewer 3JoF [2/3]**
>
> | **W3 & Q1:** Cross-Task Linearity: The theoretical analysis relies on known assumptions ... & Can the authors provide empirical analysis on the impact of the cross-task linearity assumption?  |
> | :-
>
> We appreciate the reviewers for pointing out the importance of empirical validation for our theoretical assumptions. We clarify the validity of our assumptions and provide empirical evidence below.
>
>
> ### **Empirical Verification of Cross-Task Linearity (CTL)**
> As the reviewers correctly noted, Cross-Task Linearity (CTL) is a key assumption bridging the gap between parameter-space merging and representation-space ensembling. To empirically verify how strictly this holds in our setting, **we conducted a quantitative analysis** following the reviewers' suggestions.
> We define the linear interpolation in parameter space as $\theta(\alpha) = (1-\alpha)\,\theta_i + \alpha\,\theta_j$ and in representation space as $(1-\alpha)\,f(x;\theta_i) + \alpha\,f(x;\theta_j)$. If CTL holds, the representation of the merged model $f(x;\theta(\alpha))$ should align with $(1-\alpha)\,f(x;\theta_i) + \alpha\,f(x;\theta_j)$. We measured the Cosine Similarity between them for varying $\alpha \in [0,1]$ across task pairs (taking SUN397 as an anchor).
>
> As shown in Table below, **the cosine similarity remains remarkably high across different $\alpha$ values**. This empirically demonstrates that within the shared basin of solutions derived from a common pre-trained model, the encoder's behavior is highly linear. Thus, the **CTL assumption is a robust approximation in our experimental regime**, justifying the theoretical analysis. This quantitative analysis confirms that the deviation from linearity is minimal, implying that the impact of approximation error is negligible in our setting.
>
> In the revised version, we have included this quantitative analysis and its discussion in Appendix D.4 (Table E), explicitly clarifying that it provides empirical support for the CTL assumption.
>
>
> | Coefficient | 0.1 | 0.2 | 0.3 | 0.4 | 0.5 | 0.6 | 0.7 | 0.8 | 0.9 |
> |-------------|-----|-----|-----|-----|-----|-----|-----|-----|-----|
> | Cars        | 1.00 | 0.99 | 0.98 | 0.97 | 0.97 | 0.97 | 0.98 | 0.99 | 1.00 |
> | RESISC45    | 0.99 | 0.98 | 0.97 | 0.96 | 0.95 | 0.95 | 0.96 | 0.98 | 0.99 |
> | EuroSAT     | 0.99 | 0.98 | 0.96 | 0.95 | 0.94 | 0.94 | 0.95 | 0.97 | 0.99 |
> | SVHN        | 0.99 | 0.97 | 0.94 | 0.90 | 0.87 | 0.85 | 0.87 | 0.92 | 0.97 |
> | GTSRB       | 0.99 | 0.96 | 0.93 | 0.90 | 0.88 | 0.88 | 0.90 | 0.94 | 0.98 |
> | MNIST       | 0.99 | 0.97 | 0.94 | 0.92 | 0.90 | 0.89 | 0.91 | 0.95 | 0.98 |
> | DTD         | 0.99 | 0.98 | 0.96 | 0.96 | 0.96 | 0.96 | 0.97 | 0.99 | 1.00 |

---

> ### Author Response · Authors · 2025-11-25
> **Response to the Reviewer 3JoF [3/3]**
>
> | **W3 & Q1:** The theoretical analysis relies on known assumptions ... & Are there cases where nonlinear interactions or loss non-convexity cause SyMerge to underperform? |
> | :-
>
>
>
> We would like to clarify that, under the convexity assumption, Proposition 1 holds. The proposition requires the loss function to be **convex with respect to the model output**, and in our case, this condition is satisfied by the use of cross-entropy. We have revised this point to avoid any confusions (please refer to Sec. 3.2).
>
> In practice, despite the inherent nonlinearity of neural networks, a **fine-tuned model typically remains close to its pre-trained initialization**, and the local minima reached during fine-tuning often **behave approximately like those of a convex function**. This observation **aligns with our empirical results**, including the high cosine similarity suggesting that CTL holds (as reported in the table above), and is further supported by the **strong performance of the merged model** across all experiments in our main paper.
>
> Furthermore, we acknowledge the reviewer's concern and directly address the more challenging case **where the loss surface between the two models to be merged is explicitly non-convex**. Please find our subsequent analysis below.
>
> ### **Robustness to Non-Convexity (Disjoint Basins)**
> The reviewer correctly points out that **significant parameter-space non-convexity** (e.g., models in different basins) could challenge SyMerge.
> To explicitly test this underperformance scenario, we conducted a stress test **merging models with different initializations**, where the **linearity assumption breaks down completely**.
> We merged a DTD expert ($W_1$) fine-tuned from `OpenAI-ViT` and a EuroSAT expert ($W_2$) from `DataComp-ViT`. Since no shared pre-trained weight exists, we defined task vectors relative to the weight average ($W_{avg} = (W_1 + W_2)/2$).
>
> As shown in the table below, this setting causes a **catastrophic failure for standard methods**. Weight Averaging (equivalent to utilizing $W_{avg}$) collapses, confirming the presence of a severe loss barrier. Representation Surgery also fails, as patching external adapters cannot fix the fundamentally misaligned backbone. **SyMerge, however, successfully recovers performance**. By realigning the internal layer directly, SyMerge bridges the gap even when parameter-space linearity is violated.
>
> In the revised version, we have included this stress-test experiment and its results in the paper (Section 4.3 Empirical Analyses - Merging across disjoint basins, Table 5).
>
>
> |                     | EuroSAT | DTD  | AVG  |
> |---------------------|---------|------|------|
> | Individual (Euro)   | 98.1    | 3.6  | 50.8 |
> | Individual (DTD)    | 2.0     | 79.4 | 40.7 |
> | Weight Average      | 11.6    | 2.2  | 6.9  |
> | AdaMerging          | 8.6     | 2.1  | 5.4  |
> | Surgery             | 38.1    | 13.1 | 25.6 |
> | Ours                | 96.2    | 62.0 | 79.1 |

---

> > ### Comment · Reviewer_3JoF · 2025-11-26
> >
> > Thanks to the authors for answering my questions.
> >
> > W1: The proposed Symerge method can be essentially viewed as a hybrid of surgery and AdaMerging. While the empirical results are promising, the explanation for why updating only a single layer yields superior performance is not fully convincing. The paper posits that, “updating too many layers disrupts the task-agnostic knowledge embedded in the pre-trained model, reducing its ability to generalize across tasks.” If this hypothesis is correct, it raises a logical inconsistency: shouldn't the standard fine-tuning process also be restricted to a single layer to best preserve this general knowledge?
> >
> > W3: Given that the backbone model used diverges from conventional setting, the implementation details are expected to differ significantly from standard setups. To ensure the clarity, it is crucial that the authors provide a more comprehensive description (e.g. task vector definition) of the implementation steps.

---

> > > ### Author Response · Authors · 2025-11-28
> > > **Response to the Reviewer 3JoF [1/2]**
> > >
> > > | **W1-1:** The proposed Symerge method can be essentially viewed as a hybrid of surgery and AdaMerging.  |
> > > | :-
> > >
> > >
> > > We appreciate the reviewer's follow-up question. We understand that SyMerge might appear to be a hybrid of Surgery and AdaMerging because it involves optimizing both merging coefficients and model parameters. However, we respectfully argue that SyMerge is distinct in its optimization objective and architectural target, and our new empirical results demonstrate that a **"naive hybrid" of the two baselines fundamentally fails** to achieve synergy.
> > >
> > >
> > > To address the reviewer’s hypothesis directly, we implemented and evaluated an intuitive "Hybrid Baseline" that combines the core mechanisms of both methods:
> > > - From AdaMerging: We optimized the merging coefficients using Entropy Minimization.
> > > - From Surgery: We simultaneously optimized the additional adapter modules using feature-level L1 distillation (using individual models as teachers).
> > >
> > > We tested this hybrid on the Disjoint Basins setting in W3&Q1 (EuroSAT + DTD), where standard merging fails. The results are presented below:
> > >
> > > |                     | EuroSAT | DTD  | AVG  |
> > > |---------------------|---------|------|------|
> > > | Individual (Euro)   | 98.1    | 3.6  | 50.8 |
> > > | Individual (DTD)    | 2.0     | 79.4 | 40.7 |
> > > | Weight Average      | 11.6    | 2.2  | 6.9  |
> > > | AdaMerging          | 8.6     | 2.1  | 5.4  |
> > > | Surgery             | 38.1    | 13.1 | 25.6 |
> > > | **AdaMerging + Surgery (Hybrid)**| 21.2    | 18.0 | 19.6 |
> > > | Ours                | 96.2    | 62.0 | 79.1 |
> > >
> > >
> > > The hybrid approach performs significantly worse than SyMerge. This suggests that simply combining entropy-based coefficient tuning with feature-based adapter tuning leads to optimization conflicts. **Entropy minimization often creates incorrect predictions** (Figure 4), and forcing **feature alignment on top of a misaligned backbone** (due to disjoint basins) may not recover performance.
> > >
> > > We belive that SyMerge succeeds not because it combines these methods, but because it fundamentally changes the supervisory signal and the adaptation target.
> > >
> > > - **Unlike AdaMerging's entropy minimization** (which is unsupervised and unstable) or Surgery's feature L1 loss (which is rigid), SyMerge uses the output predictions of expert models. This effectively acts as knowledge distillation, **providing a richer, more stable gradient** (verified by the high correlation with GT loss in Figure 4) that guides the joint optimization of coefficients and the layer.
> > > - **Unlike Surgery**, which adds external modules, SyMerge tunes a single internal layer without extenral ones. This allows the **shared encoder to fundamentally realign its representations** to be **compatible with the merged coefficients**, creating true synergy rather than just patching the output.

---

> > > ### Author Response · Authors · 2025-11-28
> > > **Response to the Reviewer 3JoF [2/2]**
> > >
> > > | **W1-2:** While the empirical results are promising, the explanation for why updating only a single layer yields superior performance is not fully convincing. The paper posits that, "updating too many layers disrupts the task-agnostic knowledge embedded in the pre-trained model, reducing its ability to generalize across tasks." If this hypothesis is correct, it raises a logical inconsistency:    |
> > > | :-
> > >
> > > Since the objectives of standard fine-tuning and model merging differ inherently, preserving the original features of the pretrained model often leads practical studies to fine-tune only a small subset of layers rather than the full model [1, 2]. While **full-layer fine-tuning can improve in-distribution performance** for a target task, it is well known to distort pretrained features, which in turn **reduces out-of-distribution (OOD) robustness** [1, 2]. Consequently, methods that constrain updates or **preserve pretrained weights—typically by training only a few layers** instead of the entire network—are widely used to maintain generalizability [3], aligning with our approach of updating only a few layers (i.e., a single layer).
> > >
> > > In model merging, this perspective becomes also important because **merging many tasks may correspond to handling multiple OOD scenarios**. The quality of the merged model therefore depends heavily on how well the to-be-merged features remain undistorted. This is further supported by prior work [4], which indicates that **maintaining task-specific models close to the pretrained initialization is essential for successful merging**, in line with the principle of linear mode connectivity.
> > >
> > > Empirically, our pilot study (Figure 2 in our paper) supports this view: strong cross-task performance is a prerequisite for reliable merging. Moreover, Figure 5 shows that SyMerge achieves its **best performance when adapting only a single layer**, whereas updating additional layers degrades the merged model (thus doing with less layer gives better returns as well). We attribute this to **larger updates pushing task weights out of the shared basin**, violating Cross-Task Linearity [5] and breaking the conditions needed for effective weight interpolation [4], which leads to merging failure.
> > >
> > > `References`
> > >
> > > [1] *Surgical Fine-Tuning Improves Adaptation to Distribution Shifts*, ICLR 2023
> > >
> > > [2] *Fine-Tuning can Distort Pretrained Features and Underperform Out-of-Distribution*, ICLR 2022
> > >
> > > [3] *Robust fine-tuning of zero-shot models*, CVPR 2022
> > >
> > > [4] *Linear Mode Connectivity and the Lottery Ticket Hypothesis*, ICML2020
> > >
> > > [5] *On the Emergence of Cross-Task Linearity in Pretraining-Finetuning Paradigm*, ICLM 2024
> > >
> > >
> > > | **W3:** Given that the backbone model used diverges from conventional setting, the implementation details are expected to differ significantly from standard setups. To ensure the clarity, it is crucial that the authors provide a more comprehensive description (e.g. task vector definition) of the implementation steps. |
> > > | :-
> > >
> > > We appreciate the reviewer's insightful comment and agree with the concern raised.
> > > First, the vision task experiments in the main paper are fine-tuned from the `OpenAI` pretrained weights following exactly the same procedure as in the Task Arithmetic (TA) work [1].
> > >
> > > For our additional disjoint basins experiments, we aim to verify the effectiveness of merging weights that are fine-tuned from different pretrained checkpoints for ViT-B/32. Specifically, for the DTD task we use the same fine-tuned weights as in the main paper ($W_{DTD}$), while for the EuroSAT task we do not start from the `OpenAI` pretrained weights. Instead, we use weights pretrained on the `DataComp` dataset (`datacomp_m_s128m_b4k`: a ViT-B/32 trained on a 14M subset of DataComp-1B for 128M steps with batch size 4k), and then fine-tune them on EuroSAT using the same TA-style procedure, resulting in $W_{Euro}$. The corresponding pretrained checkpoint is provided by the `open_clip` library [1].
> > >
> > > In this setting, the two tasks no longer share a common pretrained initialization $W_0$, so the standard definition of task vectors is no longer applicable. To address this, we redefine the task vectors with respect to the average of the two fine-tuned models, $W_{avg} = (W_{DTD} + W_{EuroSAT})/2,$ and then set $\tau_{DTD} = W_{DTD} - W_{avg},$ $\tau_{Euro} = W_{Euro} - W_{avg}.$
> > >
> > > We then perform merging as follows: $W_{merged}^l=W_{avg}^{l} + \lambda_{DTD}^{l} \cdot \tau_{DTD}^{l} + \lambda_{Euro}^{l} \cdot \tau_{Euro}^{l},$ where $\lambda_{DTD}$ and $\lambda_{Euro}$ denote merging coefficients for the corresponding tasks. The merging coefficients $\lambda$ are initialized to 0.3. To ensure a fair comparison, this redefinition of task vectors and the merging formulation are applied identically to all baselines. All other experimental configurations strictly follow the setup described in Appendix A.
> > >
> > >
> > > `References`
> > >
> > > [1] Editing Models with Task Arithmetic (ICLR 2023)
> > >
> > > [2] https://github.com/mlfoundations/open_clip

---

### Official Review · Reviewer_z7KY · 2025-10-31

**Soundness:** 2
**Presentation:** 2
**Contribution:** 2
**Rating:** 2
**Confidence:** 3

**Summary:**

This paper proposes a test-time adaptive model merging method, SyMerge. The key idea is to fine-tune the merging coefficients and task-specific weights through distillation between the merged model and all task experts on an unlabeled test set. Extensive experiments across vision, dense prediction, and NLP benchmarks demonstrate the effectiveness of the proposed approach.

**Strengths:**

1. This work uncovers an interesting phenomenon that stronger cross-task performance leads to better merging performance, and provides theoretical support for this observation.

2. The paper is well-organized and easy to follow.

3. The experiments demonstrate that the proposed method achieves promising results.

**Weaknesses:**

1. **Robustness to the size and quality of the unlabeled test set.** The proposed method relies on using an unlabeled test set to perform distillation between the merged model and all task experts. However, it is unclear how well this approach would work in more practical scenarios such as few-shot, long-tail, noisy, or OOD test sets. These settings naturally arise in real-world applications where users may input any query data.

2. **Potentially misleading distillation.** Each task expert is distilled on the entire test set, including samples from other tasks. For instance, when the task expert for Task A is distilled on data from Task B, its outputs may be meaningless or even incorrect. Such cross-task distillation could mislead the merged model and result in the learning of spurious or erroneous knowledge.

3. **Unclear connection between the proposed method and the core motivation (i.e., enhancing cross-task alignment).** Although the paper empirically and theoretically shows that merging performance is correlated with cross-task alignment, it remains unclear why the proposed method improves such alignment, thereby enhancing merge performance. From my perspective, the core formulation (i.e., the cross-entropy loss function) merely enforces the merged model to match the performance of each task expert, thus improving task-specific accuracy rather than directly addressing cross-task alignment. I would encourage the authors to provide more formal technical insights explaining how the proposed method and its formulation support this motivation.

4. **Limited technical novelty.** The core formulation, specifically the distillation loss, is well studied in model merging and other areas of machine learning. The self-labeling strategy appears to be a standard knowledge distillation procedure that matches the outputs between teacher and student models.

**Questions:**

See weaknesses above.

**Details Of Ethics Concerns:**

No ethics concerns.

---

> ### Author Response · Authors · 2025-11-25
> **Response to the Reviewer z7KY [1/3]**
>
> Dear Reviewer `z7KY`, The authors sincerely appreciate your thoughtful and comprehensive feedback. Your comments, including the detailed analysis of robustness concerns, potential pitfalls of cross-task distillation, the connection to cross-task alignment, and the discussion of technical novelty, have greatly helped us deepen our understanding of the strengths and limitations of our approach. We are also grateful that you acknowledged the positive aspects of this work, such as the clear organization, the interesting empirical finding on cross-task performance, and the promising experimental results. Your constructive insights were invaluable in guiding us to refine the technical clarity and motivation of the paper. We address each of your concerns in detail in the responses provided below.
>
> | **W1:** Robustness to the size and quality of the unlabeled test set |
> | :- |
>
>
> Thank you for highlighting the importance of practical deployment scenarios. First, regarding data quality (e.g., OOD, noise), **our main paper already evaluates robustness** against seven distinct types of corruption that induce domain shifts (Figure 1). While training-free methods often degrade significantly under such changes, test-time adaptive approaches, and our method in particular, maintain substantially higher performance. For example, under the strongest corruption level, **our method achieves 73.7%,** whereas the second-best method reaches 68.8%, corresponding to a **4.9%p improvement**.
>
> | Method | Task Arithmetic | LiNeS | Iso-CTS | AdaMerging | Surgery  | WEMoE | SyMerge |
> |--------|----------------:|------:|--------:|-----------:|---------------:|------:|-------:|
> | Avg.   | 55.9            | 59.4  | 65.0    | 67.2       | 68.8           | 68.6  | 73.7   |
>
>
>
>
>
> Second, to address the issue of data size and more practical and challenging scenarios, we further investigate an **'online test-time adaptation'** setting. At each iteration, the model receives **a single unlabeled test sample**, updates its parameters, and then is evaluated on that same sample (batch size = 1). So it can be regarded as a **very–few-shot scenario**. As shown in the table below, our method adapts quickly to the stream of single sample inputs and **consistently outperforms other test time adaptive approaches**. While our method can be slightly behind training-free baselines at the very beginning, its performance quickly improves as a small number of test samples arrive and then surpasses training-free methods such as Task Arithmetic, Ties-Merging, LiNeS, and Consensus TA.
>
> We have clarified in the revised version that these experiments directly address robustness to both the size and the quality of the unlabeled test data (2.2 Motivation - Limitations of training-free methods & Appendix B.5. Robustness to Data Scarcity).
>
>
>
> | SEEN SAMPLES | 10   | 20   | 50   | 100  | 200  | 500  |
> |--------------|------|------|------|------|------|------|
> | Task Arithmetic |69.1|69.1|69.1|69.1|69.1|69.1|
> | Ties Merging |72.9|72.9|72.9|72.9|72.9|72.9|
> | LiNeS |74.1|74.1|74.1|74.1|74.1|74.1|
> | Consensus TA |74.9|74.9|74.9|74.9|74.9|74.9|
> | AdaMerging   | 70.0 | 71.5 | 71.1 | 72   | 72.6 | 74.1 |
> | Surgery      | 70.0 | 71.1 | 70.9 | 73.4 | 73.2 | 73.6 |
> | **Ours**         | 73.2 | 75.3 | 75.7 | 77.1 | 80   | **82.1** |

---

> ### Author Response · Authors · 2025-11-25
> **Response to the Reviewer z7KY [2/3]**
>
> | **W2:** Potentially misleading distillation |
> | :- |
>
>
> We clarify that **our method does not distill any expert using samples from other tasks**. In SyMerge, all experts remain frozen, and each task receives supervision only from its own expert. As a result, no expert is ever optimized or influenced by data belonging to another task, and the situation where "expert A learns from task B's data" does not occur in our framework.
>
> Instead, we update only the merging coefficients that combine the task vector weights from each expert into a shared encoder. Simultaneously, the classifier for task A is trained only on the test samples of task A when applied on top of the merged encoder. In this setup, knowledge from task B affects task A only through the shared encoder representation, and **never through expert B's predictions being used as targets for task A.**
>
> Empirically, our results show that this process does not introduce spurious or misleading knowledge. Figure 4 indicates that our loss remains well aligned with the supervised loss after adaptation, and Figure 7 shows that SyMerge reduces merging-induced errors compared to prior methods, suggesting that the encoder becomes more consistent across tasks rather than absorbing noise from unrelated experts.
>
>
>
>
> | **W3:** Unclear connection between the proposed method and the core motivation (i.e., enhancing cross-task alignment) |
> | :-
>
> We appreciate the reviewer's request for a clearer connection between our formulation and cross-task alignment. In our method, each task uses its own classifier head, and we do not optimize a joint multi-task loss. Instead, the task-specific classifier and the shared merging coefficients are updated sequentially using that task's unlabeled data. The crucial mechanism is that the encoder and **merging coefficients are shared across all tasks**. As training progresses, these shared parameters must satisfy multiple task-specific objectives, which naturally **pushes the encoder toward representations that remain compatible with several task heads**. This shared optimization process is what drives improvements in cross-task alignment.
>
> Empirically, we observe this effect in multiple places. In the pilot study (Appendix B.2, Figure A), the off-diagonal entries of the cross-task performance matrix increase after adaptation, showing that cross-task alignment improves even though only task-specific losses are optimized. In Table 4, a **classifier trained with SyMerge** **can be** **attached to other merged encoders without any additional training and still improves their cross-task performance**, indicating that the learned parameters **capture structure that generalizes across tasks** rather than overfitting to a single one.
>
> To further verify this, we conducted an additional analysis: using the merging coefficients and classifier learned by SyMerge, we reconstructed each individual model $(θ^{l}_{pre} +  λ^{l}_{k} \cdot τ^{l}_{k})$ and measured its cross-task performance. We observed a substantial increase from 38.7% to 52.9%, showing that the adaptation procedure itself improves cross-task alignment even before evaluating the final merged model.

---

> ### Author Response · Authors · 2025-11-25
> **Response to the Reviewer z7KY [3/3]**
>
> | **W4:** The self-labeling strategy appears to be a standard knowledge distillation procedure |
> | :-
>
> To highlight that SyMerge offers value beyond standard KD, we present two scenarios where standard merging methods fail because the models are not compatible. In contrast, SyMerge remains robust even under these difficult conditions.
>
> ### **1. Handling Task Heterogeneity**
> Most merging methods assume tasks share a similar semantic space. However, as noted in our NLP results (Table 3), **CoLA (syntactic judgment) is qualitatively different** from other GLUE tasks (semantic understanding).
> **Prior merging methods show significant performance drops on CoLA** (Table 3), likely due to the difficulty in resolving conflicts between syntactic and semantic task vectors. Although **distillation-based** approaches like Surgery and ProbSurgery show moderate effectiveness, their performance remains **suboptimal**.
> **SyMerge successfully adapts to this heterogeneity** without needing complex conflict resolution modules, effectively bridging the gap where parameter-level merging struggles
>
> | Method   | Individual | Task Arithmetic   | Surgery | ProbSurgery | SyMerge   |
> |---------|-----------:|----------------:|--------:|-----------:|----------:|
> | `CoLA`    | 60.2       | 18.8             | 46.8    | 54.7        | **60.0**  |
>
> ### **2. Merging Models with Different Initializations**
> To test robustness against **severe parameter mismatch**, we conducted an additional experiment merging models fine-tuned from different pre-trained weights. Simple weight averaging typically yields random performance in this setting.
> We merged a DTD expert $W_1$ (from `OpenAI-ViT` init) and a EuroSAT expert $W_2$ (from `DataComp-ViT` init). Since a shared pre-trained weight does not exist, we defined task vectors relative to the weight average ($W_{avg} = (W_1 + W_2) / 2$).
> As expected, Weight Averaging and **even Surgery completely fail due to loss barrier** between these tasks. However, **SyMerge successfully recovers performance.**
>
> |                     | EuroSAT | DTD  | AVG  |
> |---------------------|---------|------|------|
> | Individual (Euro)   | 98.1    | 3.6  | 50.8 |
> | Individual (DTD)    | 2.0     | 79.4 | 40.7 |
> | Weight Average      | 11.6    | 2.2  | 6.9  |
> | AdaMerging          | 8.6     | 2.1  | 5.4  |
> | Surgery             | 38.1    | 13.1 | 25.6 |
> | Ours                | 96.2    | 62.0 | 79.1 |
>
>
> These results indicate that SyMerge is not merely applying KD; it is a **practical solution for integrating models with functional or parameter-level discrepancies** that cannot be resolved by existing merging methods.
>
> In the revised manuscript, we have (i) clarified these points about task heterogeneity and the limitations of standard KD (4.2 Main Results - NLP tasks), and (ii) incorporated the above additional experiment on merging models with different initializations (4.3 Empirical Analyses - Merging across disjoint basins, Table 5).

---

### Official Review · Reviewer_BQ5a · 2025-11-04

**Soundness:** 3
**Presentation:** 3
**Contribution:** 3
**Rating:** 8
**Confidence:** 4

**Summary:**

The paper proposes a framework for model merging, where one layer is adapted on top of doing the merge. The layer is adapted in a test-time adaptation fashion with the goal to exploit the synergies across tasks. To train the layer and find the merging coefficients they use the cross entropy loss between the output of the merged model and the corresponding expert model (single task finetuned one).

**Strengths:**

- The test of concept experiments add empirical value on the choices made on the framework
- There is a section that covers the choice of the objective function, which not only justify the selected loss but shows there was a careful experimental design process
- The experiment cover different models, showing the method works across a range of common model choices.

**Weaknesses:**

I only have minor comments, some of the figures and tables that occupy half page on pages 8 and 9 could be arranged so they do not cut the text so much like in the current version.

**Questions:**

The exploration of synergy in tasks is very interesting, however, have you consider what happens when the task being added does not play nice with the others? how well the method could minimize this interference and still get an decently performing model?

---

> ### Author Response · Authors · 2025-11-25
> **Response to the Reviewer BQ5a**
>
> Dear Reviewer `BQ5a`,The authors sincerely appreciate your thoughtful and constructive feedback. Your recognition of the strengths of our work, including the empirical value of the concept-testing experiments, the careful justification of the objective function, and the broad coverage across different model architectures, is truly encouraging. We are also grateful for your comments regarding the presentation of figures and tables, as well as your insightful question on task interference, both of which helped us reflect on important aspects for improving the clarity and robustness of our method. We provide detailed responses to your comments in the sections below.
>
> | **W1:** Arrangement of the figures and tables |
> | :- |
>
> Thank you for the constructive suggestion regarding the presentation. We have rearranged the figures and tables on pages 8 and 9 to prevent any disruption to the reading flow.
>
>
> | **Q1:** What happens when the task being added does not play nice with the others? |
> | :- |
>
> Thank you for raising this insightful question. Our main paper provides empirical evidence addressing this issue. While existing methods often fail when aggregating highly heterogeneous tasks due to interference, SyMerge is significantly more robust in such settings.
>
> For example, in the NLP experiments in Table 3 of the main paper, CoLA is qualitatively different from the other GLUE tasks: while most GLUE tasks focus on semantic judgments, **CoLA is the only syntactic task**, evaluating the grammatical acceptability of a sentence. Due to this heterogeneity, **prior merging methods yield notably poor** performance on CoLA, whereas **our method still achieves substantially higher performance**.
>
> | Method   | Individual | Task Arithmetic   | Surgery | ProbSurgery | SyMerge   |
> |---------|-----------:|----------------:|--------:|-----------:|----------:|
> | `CoLA`    | 60.2       | 18.8             | 46.8    | 54.7        | **60.0**  |
>
>
> We observe a similar pattern in our dense prediction results in Table 2. Tasks like segmentation, normal estimation, and depth estimation are **highly heterogeneous tasks**, and **standard merging methods often collapse** in this multi-task setup. In contrast, **our approach remains stable** and achieves performance close to that of the individual models even under this strong task mismatch.
>
> | Metric                     | Individual | Task Arithmetic  | Surgery  | ProbSurgery  | SyMerge          |
> |----------------------------|-----------:|------------:|--------------:|------------------:|-----------------:|
> | Segm mIoU **↑**        | 52.0       | 31.6                    | 43.3          | 43.6              | **49.8**         |
> | Depth Abs Err **↓** | 41.5       | 56.7                  | 55.3          | 52.6              | **45.3**         |
> | Normal Mean **↓**              | 24.2       | 30.6                  | 34.7          | 36.7              | **26.2**        |
>
>
> We have clarified this point and highlight these observations more explicitly in the revised version (4.2 Main Results - dense prediction tasks & NLP tasks).

---

### Author Response · Authors · 2025-11-25
**General Response**

| Key strengths of our work, as noted by the reviewers |
| :- |

We thank all the Reviewers `BQ5a`, `z7KY`, `3JoF`, `m1Pr`, `KP85` for their constructive reviews. We appreciate the positive comments regarding the novelty, theoretical grounding, and effectiveness of our method:

**Reviewer `BQ5a`** - 1) **Rigorous experimental design** justifying the framework. 2) **Versatility** across various model choices.

**Reviewer `z7KY`** - 1) **Interesting findings** supported by theoretical analysis. 2) **Promising results** with clear presentation.

**Reviewer `3JoF`** - 1) **Original perspective** shifting from non-interference to synergy. 2) **Strong theoretical backing** and **superior performance** over baselines.

**Reviewer `m1Pr`** - 1) **Novel conceptual framework**. 2) **Computational efficiency** and low implementation cost. 3) **Well-written** presentation.

**Reviewer `KP85`** - 1) **Clear motivation** and methodology. 2) **Significance** as a scalable and lightweight solution.

| Revisions of the manuscript |
| :- |

- Rearranged the figures and tables on pages 8 and 9 to improve the reading flow (**Reviewer `BQ5a`**).
- Clarified that the experiments address robustness to the quality of the unlabeled test data in Section 2.2 (**Reviewer `z7KY`**).
- Clarified the details regarding the definition of merging performance in Section 2.2 (**Reviewer `KP85`**).
- Clarified the details regarding the convexity assumption to avoid any confusion in Section 3.2 (**Reviewers `3JoF`, `m1Pr`**).
- Clarified the observations regarding task heterogeneity in NLP and dense prediction tasks in Section 4.2 (**Reviewers `BQ5a`, `z7KY`**).
- Added results demonstrating improvements beyond classifier adaptation in Section 4.3 (**Reviewer `KP85`**).
- Added stress-test experiments and results on merging models with different initializations in Section 4.3 and Table 5 (**Reviewers `z7KY`, `3JoF`, `m1Pr`**).
- Added experiments in Appendix B.5 that evaluate the model's robustness to variations in the size of the unlabeled test data (**Reviewers `z7KY`, `m1Pr`**).
- Added hyperparameter sensitivity analysis in Appendix D.3 and Figures E and F (**Reviewer `m1Pr`**).
- Added quantitative analysis and discussion providing empirical support for the CTL assumption in Appendix D.4 and Table E (**Reviewers `3JoF`, `m1Pr`, `KP85`**).
- Added experimental results on confidence-based filtering in Appendix D.5 and Table F (**Reviewers `m1Pr`, `KP85`**).
- Added analysis of computational costs in Appendix D.6 and Table G (**Reviewer `m1Pr`**).

---

### Comment · Area_Chair_dnMQ · 2025-11-27
**Request for Timely Response to Authors’ Rebuttal and Discussion**

Dear Reviewers,

I hope you are doing well. The authors have now submitted their rebuttal for the paper under your review. At this stage, your timely response is essential for ensuring a smooth discussion phase.

Could you please review the rebuttal at your earliest convenience and share your updated thoughts? If there are points that require further discussion among the reviewers, please feel free to initiate or join the conversation on the discussion thread.

Your prompt input will greatly help us maintain the review timeline. Thank you very much for your efforts and valuable contributions.

Best regards,

AC

---

### Author Response · Authors · 2025-12-01
**Summary of Rebuttal & Key Contributions for the AC [1/2]**

Dear Area Chair,

We provide this summary to facilitate an efficient review of our submission. In the following sections, we summarize the ***critical new evidence*** obtained during the rebuttal phase, which resolves key questions regarding robustness and methodology.


---

## Key Rebuttal Highlights (New Evidence)
During the rebuttal, we conducted extensive additional experiments to address reviewer inquiries. We believe these results significantly strengthen the paper's contribution and demonstrate **SyMerge's superiority over existing paradigms.**


> ### **1. Robustness to Severe Non-Convexity: Merging Across Disjoint Basins**
* Addressed concerns from **Reviewer `3JoF`** & **`KP85`** regarding the linearity assumption.
* We evaluated SyMerge in an extremely non-convex regime where models originate from **different pre-trained initializations** (`OpenAI-ViT` vs. `DataComp-ViT`). This scenario violates the standard mode connectivity assumption and exposes the failure modes of existing merging methods.
* **Result:** Standard merging (Weight Avg) and even test-time adaptive methods (AdaMerging, Surgery) completely collapse. Crucially, these baselines heavily rely on the quality of the initialized merged backbone; since the initialization fails due to the loss barrier, they cannot recover. In contrast, **SyMerge successfully restores performance** (79.1%). By actively realigning internal representations, it repairs the fundamental misalignment that coefficient tuning or external adapters cannot fix, proving its robustness even where linearity fails.


|                     | EuroSAT | DTD  | AVG  |
|---------------------|---------|------|------|
| Individual (Euro)   | 98.1    | 3.6  | 50.8 |
| Individual (DTD)    | 2.0     | 79.4 | 40.7 |
| Weight Average      | 11.6    | 2.2  | 6.9  |
| AdaMerging          | 8.6     | 2.1  | 5.4  |
| Surgery             | 38.1    | 13.1 | 25.6 |
| AdaMerging + Surgery (Hybrid)| 21.2    | 18.0 | 19.6 |
| **Ours**                | **96.2**    | **62.0** | **79.1** |




> ### **2. Methodological Distinctiveness: Why SyMerge Succeeds Where Baselines Fail**
* Addressed concerns from **Reviewers `3JoF` & `z7KY`**
* One might wonder whether SyMerge is simply a combination of `AdaMerging` (coefficient tuning) and `Representation Surgery` (adapter tuning). To test this directly, we implemented a **"Naive Hybrid Baseline"** combining AdaMerging's *entropy minimization* with Surgery’s *feature-level distillation*.
* **Result:** As shown in the table above (Point 1), the hybrid approach **collapsed (19.6%)**. This failure highlights the fundamental limitations of existing paradigms, especially in the challenging **disjoint basins** setting.
* **Why Baselines Fail:**
    * **vs. AdaMerging (Entropy Minimization):** AdaMerging relies on unsupervised entropy minimization. In severe mismatch scenarios (like disjoint basins), the model starts with high error; minimizing entropy merely forces the model to be **"confidently wrong,"** leading to collapse (5.4%).
    * **vs. Surgery (Feature Distillation):** Surgery freezes the backbone and adds external adapters. When the backbone parameters are fundamentally misaligned (disjoint), **patching external modules cannot resolve the internal feature conflicts**, resulting in poor performance (25.6%).
* **Why SyMerge Succeeds:**
    * SyMerge departs from these paradigms by using **expert-guided self-labeling** to jointly optimize the **merging coefficients** and the **internal representation itself**.
    * Unlike AdaMerging, it uses a reliable supervision signal (refer to Figure 4 in the main paper). Unlike Surgery, it **fundamentally realigns the shared encoder** instead of patching it. This enables SyMerge to bridge disjoint basins and achieve **79.1% accuracy**, recovering performance where all other paradigms fail.




> ### **3. Improvement Beyond Classifier Adaptation**
* Addressed concern from **Reviewer `KP85`**.
* One might question whether the performance gains stem solely from fine-tuning the final classifier head on the target data, rather than improving the merged model itself. To prove that our gains stem from a **better-merged encoder** (not just classifier fine-tuning), we evaluated the SyMerge-trained encoder using a **standard Zero-Shot Classifier**.
* **Result:** Even without the adapted classifier, our encoder achieves **82.4%**, significantly outperforming the Task Arithmetic initialization (**69.1%**). This confirms that SyMerge enhances the **fundamental quality of the merged representations**, independent of classifier adaptation.

---

> ### Author Response · Authors · 2025-12-01
> **Summary of Rebuttal & Key Contributions for the AC [2/2]**
>
> > ### **4. Online Test-Time Adaptation**
> * Addressed concern from **Reviewer `z7KY`**.
> * We evaluated our method in a challenging **Online TTA** setting. At each step, the model receives a single unlabeled sample (Batch Size = 1), updates its parameters on-the-fly, and is evaluated immediately.
> * **Result:** SyMerge adapts rapidly, surpassing training-free methods (Task Arithmetic, Ties-Merging) after seeing only a small number of samples, demonstrating its viability in real-time, streaming applications.
>
>
> | SEEN SAMPLES | 10   | 20   | 50   | 100  | 200  | 500  |
> |--------------|------|------|------|------|------|------|
> | Task Arithmetic |69.1|69.1|69.1|69.1|69.1|69.1|
> | Ties Merging |72.9|72.9|72.9|72.9|72.9|72.9|
> | LiNeS |74.1|74.1|74.1|74.1|74.1|74.1|
> | Consensus TA |74.9|74.9|74.9|74.9|74.9|74.9|
> | AdaMerging   | 70.0 | 71.5 | 71.1 | 72.0   | 72.6 | 74.1 |
> | Surgery      | 70.0 | 71.1 | 70.9 | 73.4 | 73.2 | 73.6 |
> | **Ours**         | **73.2** | **75.3** | **75.7** | **77.1** | **80.0**   | **82.1** |
>
>
>
>
>
>
> ---
>
> ## Other Resolved Concerns
> *  **SyMerge vs. Training-Free Methods:** Although SyMerge requires a test-time adaptation step, our experiments clearly show that the **required computation is negligible**. In online continual settings, it surpasses training-free baselines after only a few samples (see the above table for Test-Time Adaptation), indicating that meaningful gains can be achieved with minimal overhead. This small **adaptation also provides reliability in scenarios** where training-free merging becomes sensitive to task heterogeneity or disjoint basins, whereas **SyMerge remains robust**. As shown in the main paper, training-free methods degrade substantially under data corruption and distribution shifts, further highlighting the need for a small but effective adaptation step.
> * **Theoretical Validation:** We provided quantitative analysis (Cosine Similarity) confirming that the **Cross-Task Linearity (CTL)** assumption holds approximately in our experimental regime (`KP85`, `3JoF`, `m1Pr`).
> * **Task Heterogeneity:** We demonstrated robustness on the **CoLA** task (NLP), where syntactic/semantic mismatch causes other methods to fail (`BQ5a`).
> * **Scalability:** Validated on **ViT-Large** and **20-task** settings, showing consistent convergence and efficiency (`m1Pr`).
>
> ## Concluding Summary for the AC
> In summary, the new evidence directly resolves all major reviewer concerns. SyMerge **consistently outperforms existing paradigms in the most challenging settings**, including disjoint basins, online test-time adaptation, and cross-task heterogeneity. **The method is distinct, effective, and robust** across architectures and task scales. Given the strength of the new experimental evidence and the clarified methodological contributions, we believe the submission meets the bar for acceptance.

---

### Author Response · Authors · 2025-12-02
**Summary to Area Chair**

Dear Area Chair,

As the discussion period concludes, we provide a concise summary of the **distinct contributions of SyMerge**.

> ### **1. Paradigm Shift: From Non-Interference to Synergy**

We redefine the objective of model merging. While prior approaches focus on "mitigating interference" (avoiding conflicts), **we demonstrate that "Cross-Task Synergy" is essential.** We achieve this by jointly optimizing a single task-specific layer and merging coefficients, moving beyond the performance limitations of existing methods.

> ### **2. Superiority over State-of-the-Art Methods**

SyMerge achieves the best performance across Vision, Dense Prediction, and NLP tasks. Crucially, we **significantly outperform existing model-merging baselines**, including test-time adaptive methods, in complex, heterogeneous scenarios where they often struggle to maintain stability.


> ### **3. Robustness in Disjoint Basins**

We resolved robustness concerns by merging models from different initializations, a scenario where standard merging typically fails. While **baseline methods completely collapsed due to severe conflicts, SyMerge successfully recovered performance**. This proves our method works even when models are not linearly connected, overcoming a major limitation of standard task vector merging.


> ###  **4. Intrinsic Improvement of the Merged Model**

We addressed the concern that our gains might come solely from tuning the classifier. We proved that **our method fundamentally improves the merged encoder** (backbone) itself. Even without our adapted classifier, the encoder produced by SyMerge performs significantly better than standard baselines, confirming that we are improving the core representation quality.

---

We believe these distinct strengths—shifting the paradigm, outperforming SOTA methods, and verified robustness—highlight the value of SyMerge as a significant contribution to the field.

Best regards, The Authors

---

### Meta-Review · Area_Chair_MMdx · 2026-01-02

**Summary:**

This paper proposes a new model merging method that improves performance by jointly adjusting the classification heads and optimizing the merging coefficients. However, the paper still has several limitations that require further improvement. First, the motivation is not well aligned with the proposed method. Second, the technical contributions are relatively limited. Third, the proposed approach heavily relies on accurate outputs from expert models as well as the availability of target-task data, which restricts its applicability. Finally, the theoretical analysis is based on relatively strong assumptions.

**Reviewer Concerns:**

Reviewer BQ5a’s main concern is about content formatting, which can be addressed during the rebuttal period. However, Reviewer z7KY and Reviewer 3JoF mainly question the technical novelty of the paper, considering the work to be incremental. In addition, Reviewer 3JoF and Reviewer m1Pr believe that the paper relies heavily on the outputs of expert models and target test data, which limits the applicability of the proposed method. Finally, Reviewer 3JoF, Reviewer m1Pr, and Reviewer KP85 express concerns about the theoretical assumptions of the paper, arguing that the linear assumptions are unrealistic. These issues were not effectively resolved during the rebuttal period.

**Reviewer Scores:**

Reviewer BQ5a’s review provides limited information, and the main concern is about formatting. Therefore, the original score will be maintained after the rebuttal.

Reviewer z7KY’s main concerns are that the motivation of the paper does not align well with the proposed method and that the technical contribution is limited. The authors are unable to make substantial improvements to the method during the rebuttal, and thus these issues cannot be effectively addressed.

Reviewer 3JoF considers the proposed method to be an incremental extension of existing work from a technical perspective. In addition, the method relies on the outputs of expert models and test data. During the rebuttal period, Reviewer 3JoF engaged in discussion with the authors, and after the discussion, Reviewer 3JoF maintained the previous rating.

Reviewer m1Pr is mainly concerned that the proposed method relies heavily on the outputs of expert models as supervision signals, which may amplify erroneous signals. Moreover, the reliance on target data limits its usability in scenarios where target data are unavailable. Finally, the theory in the paper involves strong assumptions. These issues are difficult to resolve during the rebuttal period.

Reviewer KP85 mainly argues that the paper’s assumption of “cross-task linearity” is difficult to satisfy in practical deep neural networks.

---

### Decision · Program_Chairs · 2026-01-26

Reject